# SCALE: Causal Learning and Discovery
# of Robot Manipulation Skills using Simulation

**Tabitha Edith Lee**[*†]   **Shivam Vats**[*]   **Siddharth Girdhar**   **Oliver Kroemer**
The Robotics Institute
Carnegie Mellon University
{tabithalee, svats, sgirdhar, okroemer}@cmu.edu

**Abstract:** We propose SCALE, an approach for discovering and learning a diverse set of interpretable robot skills from a limited dataset. Rather than learning a single skill which may fail to capture all the modes in the data, we first identify the different modes via causal reasoning and learn a separate skill for each of them. Our main insight is to associate each mode with a unique set of causally relevant context variables that are discovered by performing causal interventions in simulation. This enables data partitioning based on the causal processes that generated the data, and then compressed skills that ignore the irrelevant variables can be trained. We model each robot skill as a Regional Compressed Option, which extends the options framework by associating a causal process and its relevant variables with the option. Modeled as the skill Data Generating Region, each causal process is local in nature and hence valid over only a subset of the context space. We demonstrate our approach for two representative manipulation tasks: block stacking and peg-in-hole insertion under uncertainty. Our experiments show that our approach yields diverse skills that are compact, robust to domain shifts, and suitable for sim-to-real transfer.

**Keywords:** skill discovery, causal learning, manipulation

## 1   Introduction

We want robots to help and work alongside humans in their homes, kitchens, and restaurants. However, outside of structured environments, robots currently struggle at reliably performing even some of the basic manipulation tasks that humans can do with ease. Why are humans so much better despite the vast diversity of objects and their complex interactions that they potentially need to reason about? First, humans usually know multiple ways to solve a task to be robust to failures and variations in the environment. For example, if a tight jar doesn't open with our bare hands, we may use a piece of cloth to improve our grip. Second, humans excel at selectively attending [1] to only a small part of the environment that is relevant to the task. Selective attention significantly reduces the computational complexity of reasoning and allows us to handle complicated situations.

Prior works in manipulation skill learning have leveraged these two observations separately. Most methods [2, 3, 4, 5, 6] learn skills by associating each skill with a sub-goal, where, the sub-goals are hand-designed or learned from demonstrations. Once the sub-goals have been assigned, feature selection [7, 8] and abstraction selection [9, 10] can be used to reduce the complexity of skill learning. However, such approaches are quite sensitive to the sub-goals and struggle to distinguish between different strategies to achieve the same goal. Our main insight is to *associate a skill with not just a sub-goal, but also with the variables that are causally relevant to it*. For example, opening a jar with our bare hands is a skill distinct from opening it with the help of a piece of cloth. Only *hand* and

---

[*]Equal contribution.
[†]Corresponding author.

7th Conference on Robot Learning (CoRL 2023), Atlanta, USA.

*jar* are relevant to the former, while the latter also relies on the properties of *cloth*. Hence, these two strategies should be represented as two distinct skills even though they achieve the same goal.

Based on this principle, a manipulation task involving $n$ variables can have up to $2^n$ skills, based on variable subsets being causally relevant. This is a very large space to search for skills and not all subsets may correspond to a useful skill. Hence, we propose SCALE (**S**kills from **CA**usal **LE**arning), an efficient approach for robot skill learning through causal feature selection in simulation.[1] Instead of naïvely generating data, in our approach, the robot *interacts* with the simulator by conducting causal interventions. This elicits the causal features for completing a task under different settings, yielding a diverse and compact library of skills. Our approach learns skills that are described by physically meaningful properties without spurious variables that would be related to irrelevant objects.

Our contributions of this work are two-fold. First, we introduce SCALE, an algorithm for learning a robot skill library from causal interventions in simulation. Second, we conduct a variety of experiments that demonstrate that SCALE outperforms baseline approaches for two manipulation domains of block stacking and sensorless peg insertion. As a part of these experiments, we also demonstrate sim-to-real transfer of the skills learned by SCALE for block stacking.

## 2  Related Work

**Robot skill learning.** Building robots that can solve a wide variety of complex tasks is one of the fundamental problems in robotics. A popular approach is to learn skills parameterized by the task parameters as these can generalize over related tasks. Prior works [11, 12] show such parameterized skills lie on a low dimensional piecewise-smooth manifold in the context space and identify this structure using ISOMAP [13]. For higher-dimensional problems, it becomes infeasible to learn directly in the full context space. One approach is to learn a library of simple parameterized skills which can be composed to solve more complex tasks [14, 15, 16, 17]. Recent works [18, 19] propose a differentiable attention mechanism to learn context-specific attention, but these have been evaluated only in relatively small domains. Popular methods for unsupervised skill discovery include graph-based methods [20, 21] that seek to build a graph of skills to cover the task space and information-theoretic methods [22, 23, 6] that seek to maximize the diversity of skills.

**Causality in robotics and reinforcement learning.** Causality is the science of cause and effect [24, 25, 26]. Although the advantages of causal inference and discovery within the biomedical sciences, economics, and genomics have been well-established [27, 28], the integration of causality within machine learning is nascent [29, 30]. In robotics, causality-based approaches are particularly under-explored despite the potential advantages of greater reasoning and learning capabilities [31], particularly through structure and transfer learning [32]. Most similar to our work is that of CREST [33], an algorithm for identifying features for a robot policy through causal interventions. Our algorithm SCALE leverages the work of Lee et al. on CREST for determining the causally relevant variables for each robot skill. Causality has also empowered learning the structure of physical systems from videos [34] and explanations for robot failures [35]. Within reinforcement learning more broadly, causality plays a central role for improving performance through greater structure [36], learning latent factors in dynamics via causal curiosity [37], learning invariant policies [38], and learning a dynamics model that can yield state abstractions [39].

**Intuitive physics.** Please see App. B for a discussion of how SCALE relates to intuitive physics.

## 3  Preliminaries

The robot learns a set of skills $\mathcal{K} = \{\mathcal{K}_1, \ldots, \mathcal{K}_K\}$, where each skill solves a distribution of manipulation tasks. Each task is modeled as a *manipulation MDP* [40], $M \coloneqq (\mathcal{S}, \mathcal{A}, \mathcal{R}, T, \gamma, \tau)$, where $s \in \mathcal{S}$ is the state space, $a \in \mathcal{A}$ is the action space, $\mathcal{R}$ is the reward function, $T$ is the transition

---

[1]Website: `https://sites.google.com/view/scale-causal-learn-robot-skill`

function, $\gamma$ is the discount factor, and $\tau$ is additional task information. Tasks are solved if the final reward $R_f > R_S$, where $R_S$ is a solved threshold.

**Options.** Each skill $\mathcal{K}$ is a parameterized option [41, 10]. An option $\mathcal{O} := (\pi, \mathcal{I}, \beta)$ is defined using three components: (1) the option control policy $\pi(a|s)$; (2) an initiation set $\mathcal{I} = \{R_f > R_S|s\}$ that specifies where the final reward $R_f$ solves the task when taking option $\mathcal{O}$ in state $s$ using option policy $\pi$; and (3) termination condition $\beta(s)$ that specifies when the option concludes. For the termination condition of this work, our skills execute open-loop with fixed duration.

**Context.** We define the context $c \in \mathcal{C}$ as a set of variables $\mathbf{C}$ that fully specify the manipulation task. The context space $\mathcal{C} := \mathcal{S} \times \tau$ generalizes the state space to include geometric and other time-invariant task properties defined by $\tau$. Each skill requires only a subset of the full context, determined via causal feature selection (c.f., Sec. 5.1).

**Contextual policies.** We use a hierarchical approach [42] to decompose the option control policy into an *upper-level* policy $\pi_u(\theta|c)$ and a *lower-level* policy $\pi_l(a|s, \theta)$. Given a context $c \in \mathcal{C}$, $\pi_u : c \mapsto \theta$ specifies the parameters $\theta$ for the lower-level policy $\pi_l$. For example, $\pi_l$ could be a Cartesian-space impedance controller, where $\theta$ specifies the sequence of waypoints to be followed by the robot end-effector. For our work, we assume the lower-level controller $\pi_l$ is given, and we learn the upper-level policy $\pi_{uk}$. The lower-level controller is shared across the different skills.

**Compressed Context and Feature Selection.** SCALE learns *compressed* skills that only use causally relevant context variables, as many will be unimportant. For our work, we disregard dimensions of the context space that are not chosen by causal feature selection (c.f., Sec. 5.1), leading to a compressed context space $\hat{c}$ that is obtained by selecting dimensions of the full context space that correspond to the relevant variables of interest.

**Causal Reasoning in Simulation.** SCALE leverages a simulator with the key capability of interacting with scenes through context interventions, which enables the causal learning in SCALE. For this reason, we formalize the simulator as a *causal reasoning engine* $\mathcal{W} := (\mathfrak{C}_S, T)$, where $\mathfrak{C}_S$ is the scene structural causal model (SCM) and $T$ is the transition model. App. C provides greater discussion of this formalization. This formalism addresses SCALE's assumption that the simulator is capable of answering questions to scene interventions, i.e., constructing new scenes with a change to one variable to assess if there is a change (c.f., Sec. 5.1). These variables are required to be intervenable within the simulator, but not all variables need to be intervenable. For instance, gravity is a simulation variable, but for this work, it is not considered as a candidate for causal reasoning; therefore, it does not need to be intervenable.

# 4  Skill Formulation

## 4.1  Regional Compressed Option

In our work, we formalize each robot skill $\mathcal{K}$ as a *Regional Compressed Option* (RCO), where $\mathcal{K} := (\pi_k, \text{Pre}, \beta, \mathscr{D})$ and $\pi_k$ is the option control *policy*, Pre is the *precondition*, $\beta$ is the *termination condition*, and $\mathscr{D}$ is the *data generating region* (DGR). In this model, the policy $\pi_k(a|\hat{c}_{\mathbf{A}_k})$ uses compressed context $\hat{c}_{\mathbf{A}_k}$, which is obtained by selecting dimensions of the context space according to the relevant variable set $\mathbf{A}_k \subseteq \mathbf{C}$ (c.f., Sec. 5.1). The learned, upper-level policy is $\pi_{uk}(\theta|\hat{c}_{\mathbf{A}_k})$. The precondition $\text{Pre}(c) = P(R_f > R_S|c)$ is a probabilistic initiation set [43].

## 4.2  Data Generating Region

Our goal is to learn an upper-level policy $\pi_u : c \to \theta$, i.e., a mapping that generates the correct parameters $\theta$ for solving the task from any initial context $c$. We refer to this unknown mapping as a *data generating process* or *causal process*. Instead of trying to learn this data generating process directly, which may be difficult when many variables are involved, our main insight is to model it as a mixture of multiple causal processes. Each such process is likely to have a smaller set of relevant variables and thus would be easier to learn. For example, consider the task of opening jars, where

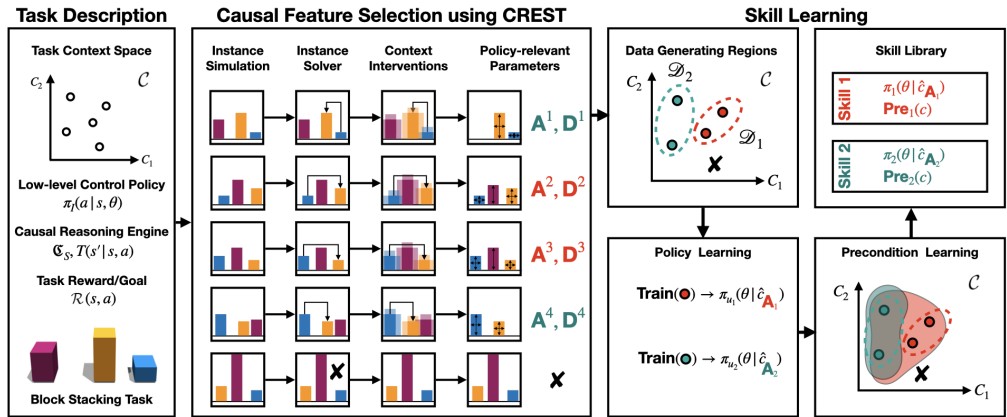

**Figure 1:** The figure shows an overview of the proposed framework applied to a block stacking task. The robot is given a context space, control policy, task simulator, and task reward. The robot samples a set of contexts to create task instances, which it subsequently solves for that instance. The robot then applies interventions on the contexts to identify skill-relevant parameters. Contexts with the same set of policy-relevant parameters come from the same causal model and are hence combined to form data generation regions. Here, we have two causal models: $\mathfrak{C}_1$ with relevant variables from the yellow, blue, and red blocks and DGR $\mathscr{D}_1$; and $\mathfrak{C}_2$ with relevant variables from the yellow and blue blocks and DGR $\mathscr{D}_2$. Each region is then used to learn a separate skill policy with the corresponding set of policy-relevant parameters. For each skill, we finally learn a set of preconditions within the context space to determine where the skill can ultimately be applied. The pairs of policies and preconditions are then combined to create a skill library for completing the given task.

the jar could be *tight* or *not tight*. We can model the data generating process for this task as a combination of two simpler causal processes: $\mathfrak{C}_1$ which uses only your hand, and $\mathfrak{C}_2$ which uses a piece of cloth along with your hand. However, these causal processes don't hold for all jar opening tasks. $\mathfrak{C}_1$ holds when the jar is *not tight*, while $\mathfrak{C}_2$ holds when the jar is *tight*. Thus, every causal process is valid only in a subset of the context space. We refer to this subspace $\mathscr{D} \subseteq \mathcal{C}$ as the data generating region (DGR) of the causal process. Here, $\mathscr{D}_1 := \{not\ tight\}$ and $\mathscr{D}_2 := \{tight\}$. The robot learns a separate skill for every such causal process. Furthermore, each skill should be trained by only using data from inside its DGR; data lying outside the DGR are generated by a different causal process and is hence out-of-distribution. The DGR uses compressed context $\hat{c}_{\mathbf{D}_k}$ obtained from relevant variable set $\mathbf{D}_k \subseteq \mathbf{C}$ (c.f., Sec. 5.1).

## 5 Skill Discovery through Causal Reasoning in Simulation

The SCALE algorithm (Fig. 1) comprises two steps: 1) skill dataset generation and 2) skill training. These steps are described in Sec. 5.1 and 5.2, respectively. Algorithm descriptions are in App. E.

### 5.1 Batch Data Generation

First, the robot interacts with the simulator $\mathcal{W}$ to collect skill training data. This is done by collecting a batch dataset $\mathcal{D}_B$. The robot samples $n$ random scenes represented by $c_i$ and attempts to determine the lower-level controller parameters $\theta_i$ that solve the specific task. In practice, we use Relative Entropy Policy Search (REPS) [44], but any suitable planner, trajectory optimizer, or reinforcement learning algorithm would suffice. Unsolved tasks are disregarded and not collected in $\mathcal{D}_B$.

**Causal feature selection.** For successfully solved scenes, the relevant variables for the policy $\mathbf{A}_i$ and DGR $\mathbf{D}_i$ are selected using the CREST algorithm [33]. CREST conducts feature selection through causal interventions. Intuitively, a variable is causal if, for all other variables held equal, interventions upon this variable induce a change in the final obtained reward $R_f$. A spurious variable has no effect on the reward and thus can safely be ignored. To summarize CREST, the process begins by solving a scene, which we refer to as the non-intervened scene. For each context variable, a new value is randomly sampled from a distribution (in the CREST work, this distribution is the context

variable's possible values). A scene is constructed with that intervened value, with all other context variables holding the same, non-intervened value. Then, the robot executes the solution to the non-intervened scene in this intervened scene to obtain an intervened reward. This process repeats a given number of times, and a statistical test is assessed to determine how often the intervened rewards differ from the non-intervened reward. If the intervened rewards are frequently no different than the non-intervened reward, the context variable is considered spurious (and causally relevant otherwise).

In this work, CREST performs interventions $\mathbf{I}$ over a local (e.g., 10%) fraction of context space $\mathcal{C}$ to yield $\mathbf{A}_i$. Similarly, $\mathbf{D}_i$ is obtained through interventions over the entire space $\mathcal{C}$. Finally, the batch dataset is appended by the dataset point $(c_i, \theta_i, \mathbf{A}_i, \mathbf{D}_i)$. Note that CREST is not a strict requirement of SCALE. In principle, SCALE requires only a determination of which variables are causally relevant, which CREST provides. Other approaches, such as using causal discovery, are also possible. An important consideration of choice of approach is whether the context space is disentangled. In our work, we assume a disentangled context space, and so the variable-by-variable intervention process of CREST (which also assumes disentangled variables) will suffice. If the context space is entangled, then causal disentanglement approaches could first be used.

**Splitting batch data into skill data.** After dataset collection, batch dataset $\mathcal{D}_B$ is split into different skill datasets according to the relevent variable sets. In this work, we assign highly occurring batch data that contain the same relevant variables $\mathbf{A}$ into the same skill dataset $\mathcal{D}_k$, while also taking the union over all associated $\mathbf{D}$. This assumption may not always hold, but is sufficient for the tasks we examine in this work. More sophisticated ways of splitting the batch dataset is left for future work.

## 5.2 Skill Training

The second phase of SCALE trains each skill $\mathcal{K}_k$ using $\mathcal{D}_k$. Each skill has relevant variable sets $\mathbf{A}_k$ and $\mathbf{D}_k$ with task solution datapoints $(c, \theta)$.

**DGR.** The DGR $\mathscr{D}$ is first trained on $\hat{c}_{\mathbf{D}_k}$ using $\mathbf{D}_k$. For this work, we use a one-class SVM to model the DGR, but in principle, any one-class classification algorithm would suffice.

**Policy.** The policy is trained next. The skill dataset contexts are filtered through the DGR $\mathscr{D}$ to obtain inliers $c^+$ for policy training data. This ensures policy training data are consistent with the underlying causal process. Then, the policy $\pi_{uk}(\theta | \hat{c}_{\mathbf{A}_k})$ is trained using $\hat{c}^+_{\mathbf{A}_k}$ (using $\mathbf{A}_k$) and the corresponding parameters $\theta^+$. For our work, policies are learned using regression, but reinforcement learning could also be used [33]. With $\pi_{uk}$ learned, the final skill policy $\pi_k(a | \hat{c}_{\mathbf{A}_k})$ is determined.

**Preconditions.** The preconditions Pre are learned last through policy evaluation. Using the simulator $\mathcal{W}$, contexts $c$ are re-sampled and evaluated with policy $\pi_k$ to obtain rewards $R_f$. This evaluation data $(c, R_f)$ is used to train a precondition classifier to obtain Pre. For our work, we use a nonlinear SVM classifier that has probability estimates.

## 6 Experimental Results

We conduct skill learning experiments with SCALE for block stacking and peg-in-hole insertion tasks with the Franka Emika Panda robot (Fig. 2). Both tasks are emblematic of high-precision control that is desirable in many industrial applications [45]. We conduct our experiments in NVIDIA IsaacGym [46, 47], a high-fidelity physics simulator that also serves as our causal reasoning engine $\mathcal{W}$. We use a custom library that implements the scene SCM $\mathfrak{C}_S$ to facilitate scene creation and interventions. The forward simulation of physics provides the transition model $T$.

**Baselines.** We compare SCALE to baseline approaches with monolithic policies (without any skills) for either the full-dimensional context space ("monopolicy") or a reduced context space obtained by using the most commonly occurring CREST result ("crest-monopolicy"). The CREST monopolicy represents naïvely using CREST, ignoring that CREST provides locally different results within the underlying data (a property that SCALE leverages).

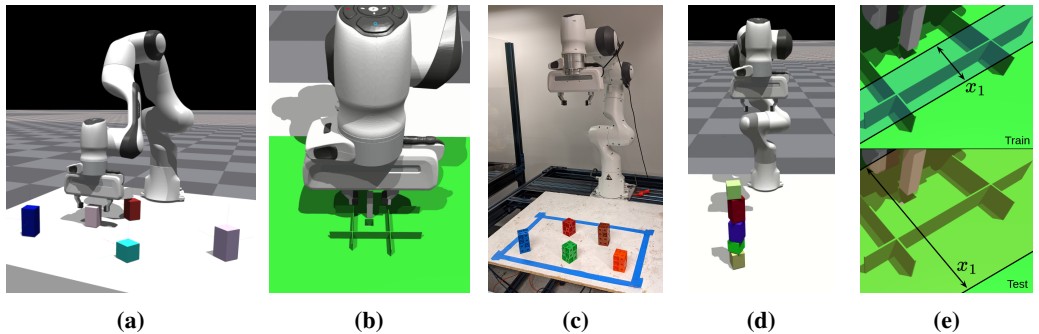

**Figure 2:** SCALE discovers skills for the Franka Emika Panda robot using causal learning in simulation for two manipulation tasks: (a) block stacking and (b) peg-in-hole insertion. In addition to skill learning experiments, we also show how SCALE can yield skills (c) for sim-to-real transfer (App. I); (d) for generalization in downstream tasks, such as stacking a block tower (App. J); and (e) for robustness to task domain shifts (App. L).

**Table 1:** Skills $\mathcal{K}_{blocks}$ that were discovered for the block stacking task. **A** and **D** are the variables used for the skill's policy and DGR, respectively. Data is the quantity of data used for each skill (from a batch dataset of 585 samples, 340 samples were used to train skills). Tsk. Sv. %, shown for both scale-lin and scale-nonlin, is the rate of task solves over the entire context space using only that skill.

| Skill | | Data | Tsk. Sv. %, Lin | Tsk. Sv. %, Nonlin. |
|---|---|---|---|---|
| $\mathcal{K}_1$ | **A**: $\{x_1^w, y_1^w, x_2^w, y_2^w\}$ 
 **D**: $\{x_1^w, y_1^w, x_2^w, y_2^w, h_2\}$ | 53 (9.06%) | 65.36% (200) | 18.36% (56) |
| $\mathcal{K}_2$ | **A**: $\{x_1^w, y_1^w, x_2^w, y_2^w, h_2\}$ 
 **D**: $\{x_1^w, y_1^w, h_1, x_2^w, y_2^w, h_2\}$ | 272 (46.50%) | 78.76% (241) | 55.88% (171) |
| $\mathcal{K}_3$ | **A**: $\{x_1^w, y_1^w, \psi_1, x_2^w, y_2^w, h_2\}$ 
 **D**: $\{x_1^w, y_1^w, \psi_1, x_2^w, y_2^w, h_2\}$ | 15 (2.56%) | 34.31% (105) | 1.31% (4) |

### 6.1 Block Stacking

**Task representation.** In the block stacking task, the robot starts with a source block ($B_1$) grasped, and it learns to place it on top of a target block ($B_2$). To do this, the robot uses a controller $\pi_l$ that defines the trajectory for the robot end-effector to traverse via impedance control. This trajectory is parameterized by $\theta_b = [\,\theta_{\Delta x},\, \theta_{\Delta y},\, \theta_{\Delta z_u}\, \theta_{\Delta z_d}\,]^{\mathrm{T}} \in \mathbb{R}^4$, which specify waypoints the robot follows sequentially. Specifically, these parameters characterize a trajectory where the robot lifts the source block vertically, moves horizontally, descends vertically, and releases the block.

For this task, the context variables $\mathbf{C}_B$ are $\{\mathbf{C}_{B_1}, \ldots, \mathbf{C}_{B_{N_B}}, h_\pi\}$, which is the union of context variables for each of $N_B = 5$ blocks plus the table height $h_\pi$ upon which the blocks are placed. The context variables for each block $b$ are $\{x_b^w, y_b^w, \psi_b, h_b, R_b, G_b, B_b\}$, yielding a 36-dimensional context space for this problem. Here, $x_b^w$ and $y_b^w$ are the world $x$- and $y$-positions of the block, and the block's orientation is represented by a rotation angle $\psi_b$ around the block's vertical axis ($z$). The $z$-dimension (height) of the block is $h_i$. Additional experimental details are available in App. H.

**Skill learning results: variable selection.** From a batch dataset of 585 samples, SCALE found the skill library $\mathcal{K}_{blocks} = \{\mathcal{K}_1, \mathcal{K}_2, \mathcal{K}_3\}$ that is shown in Tab. 1 that were learned using 340 samples of the dataset. These 340 samples were selected for being the most commonly occurring within the dataset, based on a heuristic threshold. Even though there are five blocks and 36 possible variables, the skills generally consisted of a much smaller subset of variables, relating to the geometry of the source and target blocks. Note that $\mathcal{K}_2$'s relevant variables for the policy, $\mathbf{A}_{\mathcal{K}_2}$, are consistent with earlier work by Lee et al. [33] for this domain. This is generally considered to be the "ground truth" variable result for unobstructed block motion in this case. Skill $\mathcal{K}_1$ could be seen as a version of $\mathcal{K}_2$ when $h_2$ is not needed. Rarely, the source block's rotation $\psi_1$ become important (e.g., the source block's final pose was not fully stable when stacked on the target block), and thus a skill emerges with this variable ($\mathcal{K}_3$). Variables for neither block color nor table height are observed as expected.

**Table 2:** Task evaluation results for using the skill library $\mathcal{K}_{blocks}$ for the block stacking task. Ctrl. is the approach control (skills or one monolithic policy). Fn. Cl. is the approach's function class. Linear approaches use Bayesian ridge regression, whereas nonlinear methods consist of a multilayer perceptron with a 16x16x16 architecture using ReLU activations. Task Solve % is the rate of task solves over the entire context space using the approach. Methods within $\pm 2\%$ (the stochasticity of the simulator) of the best approach are bold. $|\mathbf{A}|$ is the quantity of input variables used for the approach's policy. Data is the amount of training data used for the approach. A ground truth policy is also shown, using all context variables and additional domain knowledge.

| Approach | Ctrl. | Fn. Cl. | Task Solve % | $|\mathbf{A}|$ | Data |
|---|---|---|---|---|---|
| scale-lin (ours) | 3 skills | Linear | **90.49%** (276) | 4/5/6 | 340 |
| monopolicy-lin-all | 1 policy | Linear | 85.95% (263) | 36 | 585 |
| crest-monopolicy-lin-all | 1 policy | Linear | **89.87%** (275) | 5 | 585 |
| scale-nonlin (ours) | 3 skills | Nonlinear | **63.40%** (194) | 4/5/6 | 340 |
| monopolicy-nonlin-all | 1 policy | Nonlinear | 10.13% (31) | 36 | 585 |
| crest-monopolicy-nonlin-all | 1 policy | Nonlinear | **60.78%** (186) | 5 | 585 |
| ground-truth-policy | 1 policy | Nonlinear | 95.75% (293) | * | – |

**Skill learning results: task evaluation.** We evaluate the skill library $\mathcal{K}_{blocks}$ over the entire task distribution and show the results in Tab. 2. That is, for each context sample, the robot evaluates each skill's precondition and selects the skill with the highest probability of success. The suffix "-all" denotes that the entire batch dataset is used for the approach. For both function classes, SCALE yields an approach that outperforms full-dimensional policies and is generally comparable to CREST-reduced policies. However, the CREST-reduced policies only learn one approach to solving the task, whereas SCALE learns three. The overall best performing approach was scale-lin (90.49%) with similar performance to the CREST baseline. Performance across all nonlinear approaches was generally lower. App. H details the SCALE skill selection and further ablations.

**Sim-to-real experiment.** We transfer the skills learned by SCALE and our baselines to a real Franka Panda robot without any fine-tuning. As discussed in App. I, SCALE outperforms the baselines.

## 6.2 Sensorless Peg-in-Hole Insertion

Our second domain is peg-in-hole insertion under sensing uncertainty. It requires the robot to insert a cuboidal peg of cross-section 1 cm × 1 cm into a cuboidal hole of cross-section 1.3 cm × 1.3 cm. The robot gets a noisy initial position of the hole with the noise sampled from a Gaussian distribution $\mathcal{N}(0, 0.3^2 \text{ cm}^2)$. No further sensory observations are available. Due to this uncertainty, a naïve strategy of directly trying to push the peg down at the observed location of the hole achieves a success rate of only 34%. To address this, the robot should take *uncertainty reducing* [48] actions by initiating contact with the environment (e.g., a fixture next to the hole). Our goal in this experiment is to learn such skills autonomously.

**Task representation.** Each assembly task has 4 axis-aligned cuboidal fixtures (i.e., walls) of fixed dimensions around the hole. The 8-dimensional context variables $\mathbf{C}_P$ are $\{x_1, y_1, \ldots, x_4, y_4\}$, containing the $(x, y)$ coordinates of these fixtures with respect to the hole. The positions are different in every task, but it is always possible for the robot to localize against any of the walls to complete the task. We use a 6-parameter policy space: three $(\Delta x, \Delta y, \Delta z)$ actions executed in sequence in the robot's end-effector frame. In every policy, $\Delta z$'s are designed to move the peg down while $\Delta x$ and $\Delta y$ are parameters that are learnt using RL. Additional experimental details are available in App. K.

**Skill learning results: variable selection.** Table 3 enumerates the skills $\mathcal{K}_{peg} = \{\mathcal{K}_1, \ldots, \mathcal{K}_5\}$ discovered by SCALE. Skills $\mathcal{K}_{2-5}$ localize against one of the 4 walls. For each such skill, only the wall being used for localization is relevant to the skill and the other walls can be ignored. Consequently, the set of relevant variables for these skills contains only the distance to the wall that it localizes against. For the linear case, all of these skills except for $\mathcal{K}_3$ have high success rate, whereas success rates are slightly lower for the nonlinear approach. Interestingly, SCALE also discovers a skill $\mathcal{K}_1$ that has an empty set of relevant variables. For more discussion of this skill, see App. K.

**Table 3:** Skills $\mathcal{K}_{peg}$ that were discovered for the peg-in-hole insertion task. Columns are the same as in Tab. 1, except Data represents which 168 samples were used to train skills (from a batch dataset of 210 samples).

| Skill | A | D | Data | Task Solve %, Lin | Task Solve %, Nonlin. |
|---|---|---|---|---|---|
| $\mathcal{K}_1$ | $\{\}$ | $\{x_1, y_2, y_3, x_4\}$ | 56 (26.67%) | 64.84% (166) | 61.72% (158) |
| $\mathcal{K}_2$ | $\{x_4\}$ | $\{x_4\}$ | 25 (11.90%) | 97.66% (250) | 84.38% (216) |
| $\mathcal{K}_3$ | $\{x_1\}$ | $\{x_1\}$ | 27 (12.86%) | 44.53% (114) | 84.77% (217) |
| $\mathcal{K}_4$ | $\{y_3\}$ | $\{y_3\}$ | 28 (13.33%) | 94.14% (241) | 82.81% (212) |
| $\mathcal{K}_5$ | $\{y_2\}$ | $\{y_2\}$ | 32 (15.24%) | 98.44% (252) | 79.69% (204) |

**Table 4:** Task evaluation results for using the skill library $\mathcal{K}_{peg}$ for peg insertion (columns in Tab. 2).

| Approach | Ctrl. | Fn. Cl. | Task Solve % | \|A\| | Data |
|---|---|---|---|---|---|
| scale-lin (ours) | 5 skills | Linear | **96.48%** (247) | 0/1/1/1/1 | 168 |
| monopolicy-lin-all | 1 policy | Linear | 62.50% (160) | 8 | 210 |
| crest-monopolicy-lin-all | 1 policy | Linear | 62.89% (161) | 1 | 210 |
| scale-nonlin (ours) | 5 skills | Nonlinear | **88.67%** (227) | 0/1/1/1/1 | 168 |
| monopolicy-nonlin-all | 1 policy | Nonlinear | 12.89% (33) | 8 | 210 |
| crest-monopolicy-nonlin-all | 1 policy | Nonlinear | 55.47% (142) | 1 | 210 |

**Skill learning results: task evaluation.** Table 4 presents the task evaluation of the skill library $\mathcal{K}_{peg}$ for 256 randomly sampled tasks. For both linear and nonlinear cases, SCALE outperforms both baselines. The low success of monopolicy-nonlin-all is likely due to insufficient data owing to a larger network. The most common CREST result was variable $x_4$ (21.90%), so this was used for the CREST baselines. However, it only localizes against one wall. The improvement of SCALE over the CREST baselines implies SCALE skills benefit from the DGRs through greater quality training data, whereas the CREST approaches use the entire dataset despite most samples having a differing CREST result than $x_4$. For details of SCALE skill selection and further ablations, see App. K.

**Domain shift experiment.** To evaluate the out-of-distribution generalization capabilities of SCALE, we evaluate the skills on a test distribution that is significantly harder than the training distribution. All approaches see a degradation in performance, but ours is more robust. See App. L for details.

## 7 Conclusion

We present SCALE, an approach for discovery of compact, diverse robot manipulation skills from causal interventions in simulation. These skills arise from the skill DGR: a region that captures the underlying data generating process. We demonstrate the advantages of skill libraries discovered with SCALE for two simulation domains as well as on a real robot system.

**Limitations and future work.** SCALE assumes the robot has access to a causal reasoning engine. We provide this via simulation and scene structural causal models, but these models could be learned via causal discovery. SCALE primarily learns from batch dataset collection; active learning of skills would reveal useful behaviors that are statistically uncommon in the batch setting. Lastly, SCALE assumes that the context variables are defined, intervenable, and disentangled. For tasks and domains where these assumptions do not currently hold, future work in adjacent fields may ultimately provide a path forward. Specifically, causal representation learning [29] — learning high-level intervenable variables from low-level observations — could construct a state representation that SCALE can use, and, if a representation is available but entangled, causal disentanglement could be used.

### Acknowledgments

We gratefully acknowledge support from the National Science Foundation (Grant No. CMMI-1925130), U.S. Office of Naval Research (Grant No. N00014-18-1-2775), U.S. Army Research Laboratory (Grant No. W911NF-18-2-0218 as part of the A2I2 Program), and the NVIDIA NVAIL Program. We also gratefully thank our reviewers, whose helpful comments strengthened this work.

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

## A  SCALE and Appendices Overview

Fundamentally, SCALE is a causal learning algorithm for discovering compact, diverse skills through interventions in simulation. Figure 3 provides an overview of the approach.

**Structure of appendices.** These appendices are structured as follows. Appendix B describes how SCALE connects to related work in intuitive physics. Appendix C provides greater details into the formalization of the simulator and its role as a causal reasoning engine. Appendix E formalizes the SCALE algorithm using nomenclature introduced in App. D. A discussion of higher-dimensional context spaces and SCALE is then provided in App. F. Next, App. G provides a toy experiment that is designed to convey greater intuition and visualization of the mechanisms that underlie SCALE. Appendix H presents additional experimental details of the block stacking experiment presented in Sec. 6.1. Following this, Apps. I and J provides two additional experiments in the block stacking domain: a sim-to-real transfer experiment and a downstream task evaluation experiment, respectively. The next two appendices concern the peg-in-hole insertion domain. Appendix K details additional experimental details first presented in Sec. 6.2, and App. L presents an additional experiment that shows the robustness of SCALE under a task domain shift. Lastly, Appendix M contains a primer on causality for readers who are new to this area of research.

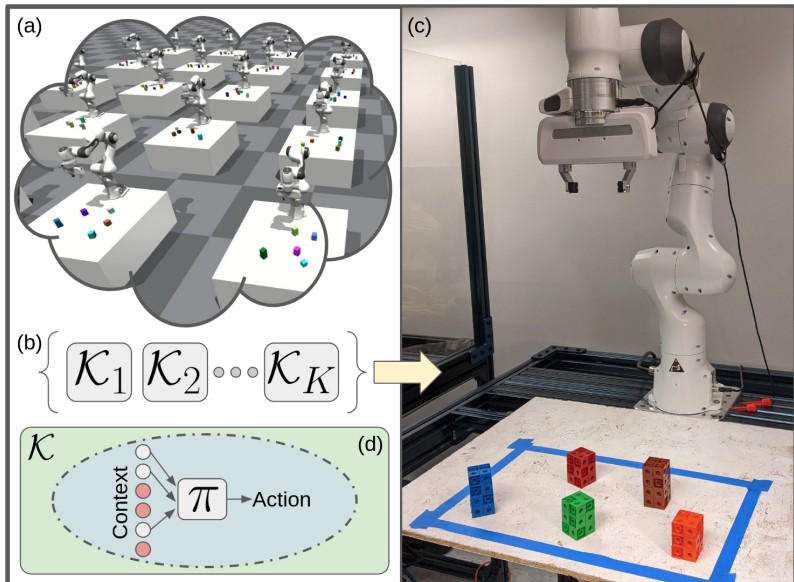

**Figure 3:** In SCALE, the robot discovers skills in simulation using causal learning. (a) The simulation is used to solve task instances and conduct interventions to determine causally relevant context variables. (b) Simulation data are used to train a library of skills, (c) which are suitable for sim-to-real transfer learning. (d) Each skill that is learned is parameterized by the relevant variables selected in simulation. Here, red context variables are unnecessary for the skill policy and can be safely ignored. The boundary encircling the policy represents the skill DGR and precondition, which are also learned.

## B  Related Work for Intuitive Physics

This appendix describes the connections between SCALE and the intuitive physics literature. Intuitive physics is the ability to approximately predict and model the physical world without explicit understanding of the underlying dynamics [49]. Literature in cognitive psychology has suggested that humans develop mental intuitive physics models to support fast prediction and understanding of complex physical scenes which enables physical reasoning [50]. Computational learning of intuitive physics have been successful, enabling reinforcement learning and planning applications owing to the models ability for forward prediction [51, 52, 53]. In our work, our causal reasoning engine can

be viewed as an internal model that uses interventions to elicit the physical mechanisms by which the data arise.

## C  Simulation as a Causal Reasoning Engine

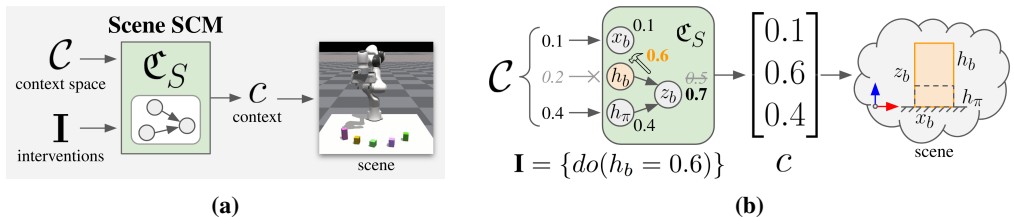

**(a)**                                           **(b)**

**Figure 4:** Illustrations of the scene structural causal model used in the simulator $\mathcal{W}$. (a) From context space $\mathcal{C}$ and robot interventions $\mathbf{I}$, the scene SCM $\mathfrak{C}_S$ generates a context vector $c$ that represents a particular *scene* that defines objects and their properties. (b) In this block example, $\mathfrak{C}_S$ is defined using scene variables $\boldsymbol{\Psi} := \mathbf{C} \cup z_b$ and context variables $\mathbf{C} := \{x_b, h_b, h_\pi\}$, where $x_b$ is block x-position, $h_b$ is block height, $h_\pi$ is table height upon which the block rests, and $z_b := \frac{1}{2}h_b + h_\pi$ is block z-position. Normally, values of $\mathbf{C}$ are sampled from context space $\mathcal{C}$, but the robot performs an intervention $\mathbf{I} = \{do(h_b = 0.6)\}$ to force the value of $h_b$ to be 0.6. As a result, the dependent variable $z_b$ is determined as 0.7 using this intervened value. Lastly, the scene is constructed and represented as context vector $c = [0.1, 0.6, 0.4]^\mathrm{T}$.

This appendix provides greater discussion of the simulator formalization used by SCALE. The simulator model, $\mathcal{W} := (\mathfrak{C}_S, T)$, is formalized as follows:

1. a scene structural causal model $\mathfrak{C}_S$ (Fig. 4) that, given context space $\mathcal{C}$ and interventions $\mathbf{I}$, instantiates a scene that can be represented as a context vector, $c \in \mathcal{C}$;

2. the transition model $T$ that captures the domain forward dynamics as the robot interacts with the world through $\theta$ starting from the scene initialized from $\mathfrak{C}_S$.

A structural causal model (SCM) [25, 26] can be represented as a directed acyclic graph that is driven by exogenous variables (functional inputs of the graph) that produces the solution for all variables within the graph. These two components of the simulator capture the spatial structure inherent to the scene itself ($\mathfrak{C}_S$), and the spatiotemporal structure of the robot interacting with the world ($T$). The simulator model $\mathcal{W}$, including the scene SCM and transition function, is provided for the robot to use. In principle, the scene SCM could be learned via causal representation learning [29], e.g., a world models approach that admits causal interventions.

The scene SCM $\mathfrak{C}_S$ is defined by structural equations with scene variables $\boldsymbol{\Psi}$, where $\mathbf{C} \subseteq \boldsymbol{\Psi}$. In the graph induced by $\mathfrak{C}_S$, the scene variables are the nodes, and context variables $\mathbf{C}$ are the root nodes and exogenous variables (functional inputs) of the SCM. The value of the context variables is given by interventions $\mathbf{I} = \{do(C_i = c_i)\}$ if specified, or otherwise sampled from the context space $\mathcal{C}$. The robot only conducts interventions with respect to $\mathbf{C}$ that would yield a steady-state solution and are physically realizable, excluding physically invalid scenes (e.g., object penetration).

The transition model $T$ is the same as typical simulators. The forward dynamics are simulated through the initial state $s_0$, obtained from the scene created by $\mathfrak{C}_S$, and $\theta$, the inputs to the low-level controller $\pi_l$. With these inputs, the system temporally evolves as usual until the end of the episode, where reward $R_f$ is obtained and compared to a threshold $R_S$ to determine if the task was solved.

## D  Nomenclature

Table 5 summarizes the nomenclature used in this paper and, in particular, the SCALE algorithm (c.f., App. E). Note the use of italics and bold type to disambiguate certain symbols. For example,

**Table 5:** Table of nomenclature.

| Symbol | Meaning |
|---|---|
| $\mathbf{X}$ | set of $d$ random variables, i.e., $\mathbf{X} := \{X_1, \ldots, X_d\}$ |
| $\mathcal{X}$ | space of $\mathbf{X}$, i.e., $\mathcal{X} := [\mathcal{X}_1, \ldots, \mathcal{X}_d]^{\mathrm{T}}$ |
| $x$ | vector instantiation of $\mathbf{X}$ i.e., $x := [x_1 \in X_1 \subseteq \mathcal{X}_1, \ldots, x_d \in X_d \subseteq \mathcal{X}_d]^{\mathrm{T}}$ |
| $\mathcal{K}$ | set of $k$ robot skills, i.e., $\mathcal{K} := \{\mathcal{K}_1, \ldots, \mathcal{K}_k\}$ |
| $\mathcal{D}$ | dataset containing $^{\mathbf{A}}X \in \mathbb{R}^{m \times n}$ samples from set $\mathbf{A}$ with size $n$ and $^{\mathbf{B}}Y \in \mathbb{R}^{m \times p}$ labels from set $\mathbf{B}$ with size $p$ |

$\mathbf{X}$ is a set of random variables, but $X$ refers to a dataset matrix. The notation for a variable and its instantiation as a scalar may also be overloaded depending on the context.

## E   SCALE Algorithm

As explained in Sec. 5, the SCALE algorithm (Alg. 1) describes how the skills are learned through batch dataset collection and skill training. The procedure for batch dataset collection used by SCALE (SKILLTRAINDATA) is described in Alg. 2.

Note that the number of skills is not a hyperparameter of the SCALE algorithm. Rather, the skill quantity emerges from SPLITINTOSKILLDATASETS from groups of highly-occurring CREST results, where each group becomes the dataset for a particular skill.

## F   SCALE and Higher-Dimensional Context Spaces

The SCALE algorithm scales linearly with the dimensionality of the context space, i.e., $O(|\mathbf{C}|)$, due to the necessity of performing interventions on each context variable. In the experiments examined in this work, the dimensionality of the context space was 36 and 8 for the block stacking and peg insertion domains, respectively. For other applications where the context space is very large, heuristics can be incorporated to first downselect the context space into a smaller candidate space that can be provided to SCALE. Example heuristics could include a distance metric (objects closer to the goal may be more likely to be relevant than those further away) or using other approaches such as meta-level priors [54].

## G   Block Stacking Intuitive Example

To provide greater intuition for SCALE and the causal skill learning problem, we present the *Height-Height* experiment (Fig. 5): a simple example in the block stacking domain that can be easily visualized.

**Task and policy description.** The *Height-Height* experiment contains 3 blocks: 1) a source block; 2) a target block; and 3) an obstructing block between the source and target block. As in Sec. 6.1, the task is to place the source block on top of the target block. The same controller is used as in Sec. 6.1, which is parameterized by $\theta \in \mathbb{R}^4$. Specifically, each parameter of the controller is defined as follows:

1. $\theta_{\triangle x}$: the distance the source block is moved along the world coordinate frame's $+x$-axis once it is picked up.

2. $\theta_{\triangle y}$: the distance the source block is moved along the world coordinate frame's $+y$-axis once it is picked up.

**Algorithm 1:** SCALE: SKILLS FROM CAUSAL LEARNING

---

**Input:** causal reasoning engine $\mathcal{W}$, context space $\mathcal{C}$, controller $\pi_l$, reward solved threshold $R_S$, number of samples $n$, skill policy function $f_\pi$, number of evaluations $m$, skill timestep $T_f$

**Initialize:** skills $\mathcal{K} \leftarrow \emptyset$

---

```
// Collect training data
```
$(\mathcal{D}_1, \ldots, \mathcal{D}_k) \leftarrow \text{SKILLTRAINDATA}(\mathcal{W}, \mathcal{C}, \pi_l, n)$
```
// Train skills
```
**for** $j = 1$ **to** $k$ **do**

    $({}^C\boldsymbol{X}, {}^\theta\boldsymbol{Y}, \mathbf{A}, \mathbf{D}) \leftarrow \mathcal{D}_j$
```
    // Train DGR
```
    ${}^D\boldsymbol{X} \leftarrow \text{REDUCEDIMS}({}^C\boldsymbol{X}, \mathbf{D})$
    $\mathscr{D} \leftarrow \text{TRAINDGR}({}^D\boldsymbol{X})$
```
    // Train Policy
```
    ${}^A\boldsymbol{X} \leftarrow \text{REDUCEDIMS}({}^C\boldsymbol{X}, \mathbf{A})$
    $({}^A\boldsymbol{X}^+, {}^\theta\boldsymbol{Y}^+) \leftarrow \text{DGRINLIERS}(\mathscr{D}, {}^A\boldsymbol{X}, {}^D\boldsymbol{X}, {}^\theta\boldsymbol{Y})$
    $\pi_u \leftarrow \text{TRAINPOLICY}(f_\pi, {}^A\boldsymbol{X}^+, {}^\theta\boldsymbol{Y}^+)$
    $\pi \leftarrow \pi_l \pi_u$
```
    // Train Preconditions
```
    $({}^C\boldsymbol{X}_e, {}^R Y_e) \leftarrow \text{EVALUATEPOLICY}(\mathcal{W}, \mathcal{C}, \pi, m)$
    $\text{Pre} \leftarrow \text{TRAINPRECONDITION}({}^C\boldsymbol{X}_e, {}^R Y_e, R_S)$
```
    // Set Termination Conditions
```
    $\beta \leftarrow T_f$
```
    // Construct Skill
```
    $\mathcal{K} \xleftarrow{+} (\pi, \text{Pre}, \beta, \mathscr{D})$

**end**

---

**Result:** learned skills $\mathcal{K}$

---

    3. $\theta_{\Delta z_u}$: the distance the source block is lifted (moved along the world coordinate frame's $+z$-axis) during the pick-up motion.

    4. $\theta_{\Delta z_d}$: the distance the source block descends (moved along the world coordinate frame's $-z$-axis) during the set-down motion.

The controller behaves as follows:

    1. Move robot end-effector to source block and grasp it.

    2. Lift up the source block according to $\theta_{\Delta z_u}$.

    3. Move the source block in the $x$-$y$ plane according policy parameters $\theta_{\Delta x}$ and $\theta_{\Delta y}$.

    4. Set down the source block according to $\theta_{\Delta z_d}$.

    5. Ungrasp the source block.

The context space of this experiment is just 2 variables, $h_t$ and $h_o$, facilitating 2-dimensional visualizations. For greater clarity, we refer to block properties by whether they belong to the target block ($t$) or the obstructing block ($o$), instead of their index (as in Sec. 6.1). For this experiment, only linear approaches are considered.

**Skill learning results.** The SCALE results for the *Height-Height* experiment are shown in Tab. 6 and Fig. 6. The dataset size for skill learning was 569 samples, from an original size of 581. The remaining 12 samples consisted of CREST results that occurred rarely (2.07%), and thus they were not used for skill learning. Additionally, Fig. 7 visualizes the policy parameters of the dataset. Two primary behaviors were learned: *free motion* ($\mathcal{K}_{free}$), and *obstructed motion* ($\mathcal{K}_{obstr}$). These behaviors emerge *because of* the causal relationships between context variables.

**Algorithm 2:** SKILLTRAINDATA

**Input:** causal reasoning engine $\mathcal{W}$, context space $\mathcal{C}$, controller $\pi_l$, reward solved threshold $R_S$, number of samples $n$, local region fraction $f$, minimum dataset size $d$

**Initialize:** batch dataset $\mathcal{D}_B \leftarrow \emptyset$

```
// Collect training data
```
**for** $i = 1$ **to** $n$ **do**
    $c \leftarrow$ SAMPLEVALIDSCENE($\mathcal{W}, \mathcal{C}$)
    $(\theta, R_f) \leftarrow$ TRYTOSOLVETASK($\mathcal{W}, c, \pi_l$)
    $TaskSolved \leftarrow R_f > R_S$
    **if** $TaskSolved$ **then**
        $\mathbf{A} \leftarrow$ CREST($\mathcal{W}, c, \pi_l, \theta, R_f, f\mathcal{C}$)
        $\mathbf{D} \leftarrow$ CREST($\mathcal{W}, c, \pi_l, \theta, R_f, \mathcal{C}$)
        $\mathcal{D}_B \xleftarrow{+} (c, \theta, \mathbf{A}, \mathbf{D})$
    **end**
**end**
```
// Separate into k skill datasets
```
$(\mathcal{D}_1, \ldots, \mathcal{D}_k) \leftarrow$ SPLITINTOSKILLDATASETS($\mathcal{D}_B, d$)

**Result:** skill training data $(\mathcal{D}_1, \ldots, \mathcal{D}_k)$

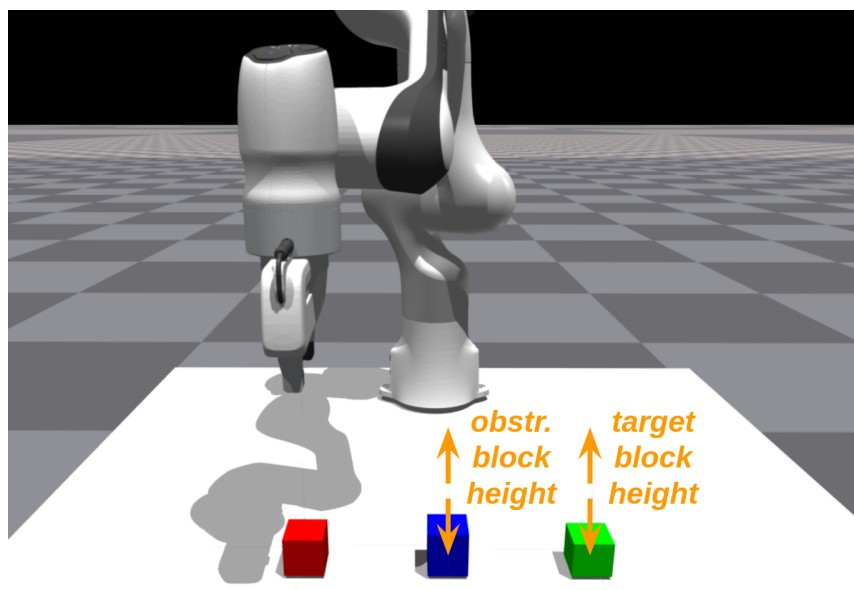

**Figure 5:** The *Height-Height* experiment is an intuitive example for SCALE in the block stacking domain. In this experiment, only two context variables can vary: the height ($z$-dimension) of the obstructing block ($h_o$) and the height of the target block ($h_t$). All others variables (e.g., features of the source block) do not change throughout this experiment.

When the obstructing block is shorter than the target block (i.e., $h_t > h_o$), then the obstructing block height can safely be ignored in the robot action (thus, $h_o \nsubseteq \mathbf{A}$ for $\mathcal{K}_{free}$). This is reflected by the values of $\theta_{\Delta z_u}$ and $\theta_{\Delta z_d}$ in Fig. 7. In the region corresponding to $\mathcal{K}_{free}$, $\theta_{\Delta z_u}$ varies linearly with respect to the target block height, but not with the obstructing block height. Thus, $\theta_{\Delta z_d}$ is generally $0$. The result is that the robot tends to lift the block to a value that depends on the target block height, and no set-down motion ($\theta_{\Delta z_d}$) is needed.

However, when the obstructing block is taller than the target block (i.e., $h_t < h_o$), the obstructing block's geometry interferes with the robot's motion, and the robot must take this into account when

taking action. Specifically, the robot must first lift the source block over the obstructing block. After it moves laterally, the robot must descend to set the source block down; dropping the block would typically lead to inadequate reward to solve the task. Because both the heights of these blocks are needed to perform this action, $\{h_t, h_o\} \subseteq \mathbf{A}$ for $\mathcal{K}_{obstr}$. In Fig. 7, the effect of $h_o$ appears in the $\theta_{\Delta z_u}$ parameter values, where the variation in the $\mathcal{K}_{obstr}$ region arises because of needing to lift above the obstructing block height, $h_o$ (and thus, this parameter no longer depends on $h_t$). However, for $\theta_{\Delta z_d}$, *both* $h_t$ and $h_o$ are needed, as the distances the robot descends through $\theta_{\Delta z_d}$ arises from the difference between $h_t$ and $h_o$. Thus, the gradient here shows components for both $h_t$ and $h_o$.

These two skills encode the two distinct data generating processes within this context space. These processes — the reason why the data are generated a certain way — fundamentally depend on whether the obstructing block is shorter or taller than the target block. Whether a condition holds for a given context requires the value of both of the blocks heights, so both block heights are needed to define each skill's data generating region (i.e., $\{h_t, h_o\} \subseteq \mathbf{D}$).

Note that neither skill can robustly solve the entire task space (55.63% for $\mathcal{K}_{free}$ and 57.50% for $\mathcal{K}_{obstr}$). However, when using the entire library $\boldsymbol{\mathcal{K}}_{HH} = \{\mathcal{K}_{free}, \mathcal{K}_{obstr}\}$ (Tab. 7), the success rate becomes 100.00%, with each skill being selected at approximately 50% chance (49.38% for $\mathcal{K}_{free}$, and 50.62% for $\mathcal{K}_{obstr}$). This is expected because the relationship $h_t > h_o$ holds for half of the context space and $\mathcal{K}_{free}$ should be used, whereas $h_t < h_o$ ($\mathcal{K}_{obstr}$) holds for the other half.

**Table 6:** Skills $\boldsymbol{\mathcal{K}}_{HH}$ that were discovered for the *Height-Height* experiment. $\mathbf{A}$ and $\mathbf{D}$ are the variables used for the skill's policy and DGR, respectively. Data is the quantity of data used for each skill (from a batch dataset of 581 samples, 569 samples were used to train skills). These samples are used to train a linear policy (Bayesian ridge regression) using the features from variables in $\mathbf{A}$. Task Solve % is the rate of task solves over the entire context space using only that skill.

| Skill | $\mathbf{A}$ | $\mathbf{D}$ | Data | Task Solve % |
|---|---|---|---|---|
| $\mathcal{K}_{free}$ | $\{h_t\}$ | $\{h_t, h_o\}$ | 253 (43.55%) | 55.63% (178) |
| $\mathcal{K}_{obstr}$ | $\{h_t, h_o\}$ | $\{h_t, h_o\}$ | 316 (54.39%) | 57.50% (184) |

**Baseline comparisons.** In addition to scale-lin, Tab. 7 shows comparisons against several baselines. The "monopolicy" baselines are monolithic policies (without skills). The "-sk" and "-all" suffixes denote whether the monolithic policy uses the same data as the SCALE library ("-sk", 569 samples) or the entire batch dataset ("-all", 581 samples). Given the similar amount of data, it is unsurprising that monopolicy-lin-sk and monopolicy-lin-all are essentially the same up to the stochasticity of the simulator ($\pm 2\%$). Note that, unlike in Sec. 6.1 and Sec. 6.2, CREST monopolicy baselines are not examined in this experiment; they are functionally equivalent to the monopolicy approaches because the most common CREST result is $\{h_t, h_o\}$, which is the same as the entire context space used for the monopolicy baselines.

As shown in Tab. 7, the skill library obtained by SCALE vastly outperforms the baselines, providing task evaluation performance similar to that of a ground truth policy. This outcome is possible because SCALE learns underlying regions of similar causal structure within the data, whereas monolithic policies ignore such structure. As shown in Fig. 7c–7d, this domain is nonlinear, but can be represented by two smaller linear regions ($h_t > h_o$ and $h_t < h_o$). Learning to regress to both regions with a monolithic linear policy is not possible, but SCALE can solve this domain with separate linear skills, one per region.

**Summary.** Our approach for SCALE — learning skills that encode distinct causal processes — empowers the robot with a diversity of specialized behaviors to use, depending on the context. Generalization of the context space can be achieved then through the composition of these behaviors, rather than attempting to learn a monolithic skill or policy that can capture the entire variation. In this example, two skills each with a linear policy is sufficient for generalization with SCALE, whereas a monolithic approach would require a nonlinear policy.

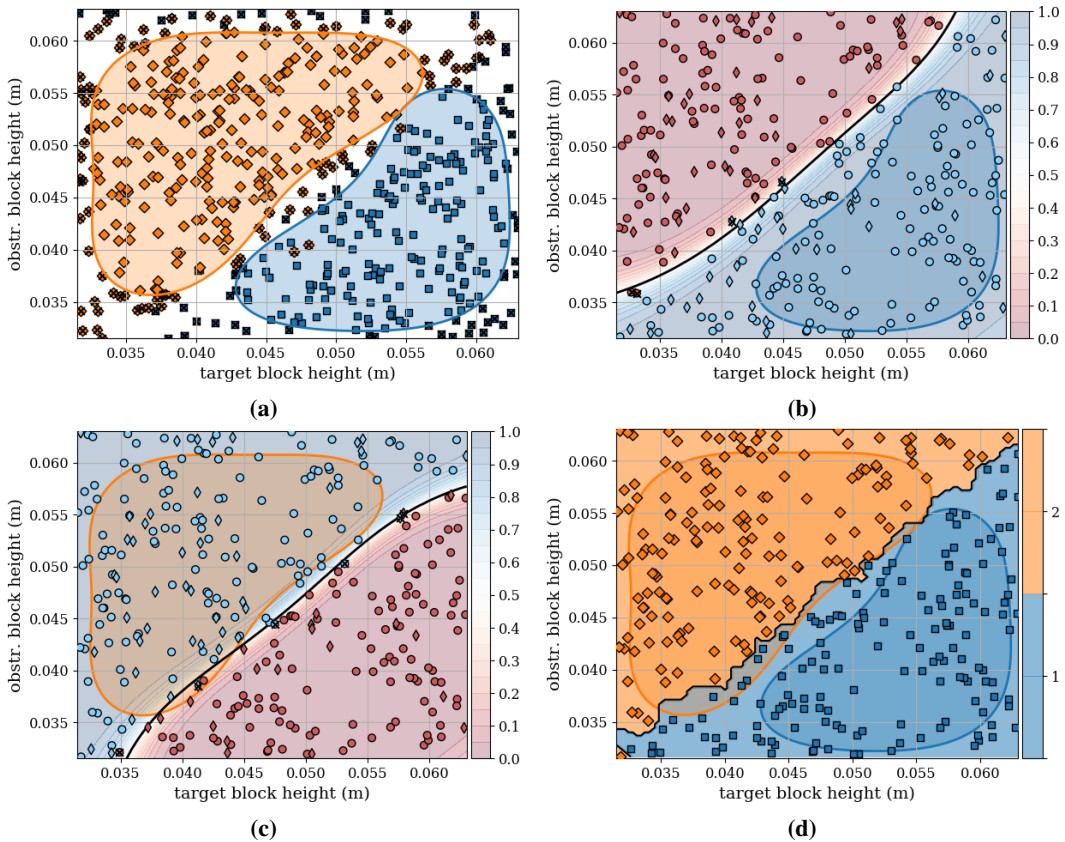

**Figure 6:** SCALE results for the *Height-Height* experiment. Two skills were found: $\mathcal{K}_{free}$ (free block motion), stylized in blue with rectangular markers, and $\mathcal{K}_{obstr}$ (obstructed block motion), stylized in orange with diamond markers. (a) Learned data generating regions. Each datapoint is a result from CREST. Datapoints that are crossed out are considered outliers and not used for training the policy for that skill. (b–c) Preconditions for $\mathcal{K}_{free}$ and $\mathcal{K}_{obstr}$, respectively. The black line is the decision boundary for the prediction of whether the task would or would not be solved with that skill. Note that each skill's DGR generally falls within the positive precondition boundary. Training and test data for learning the preconditions are indicated by circle and thin diamond markers, respectively. Datapoints that result in a different prediction than observed are crossed out. (d) Task evaluation when using the skill library $\{\mathcal{K}_{free}, \mathcal{K}_{obstr}\}$ to solve the task. The marker and color of each datapoint indicate which skill was selected for completing the task based on the skill preconditions (i.e., the skill with the highest probability of success). Note that the separation between selecting $\mathcal{K}_{free}$ and $\mathcal{K}_{obstr}$ is consistent with each skills' underlying precondition and DGR. Datapoints that were not solved by the chosen skill are crossed out.

## H   Additional Details for Block Stacking Experiment

This appendix provides greater information for the block stacking experiment first presented in Sec. 6.1.

**Context.** Note that the block vertical position $z_b^w \in \boldsymbol{\Psi}$ is not part of the context, as we only consider cases where the scene can be initialized into a steady state condition. Thus, $z_b^w := \frac{1}{2}h_b + h_\pi$.

**Reward function.** The reward function for the task is $R = R_B - \alpha_L L - \alpha_e e - \alpha_d d$, where $R_B = 10$ is a bonus term obtained when the block is successfully stacked, $L$ is the total end-effector path of the robot ($\alpha_L = 1$), $e$ is the L2 norm error between the source block at the time of release and the goal ($\alpha_e = 1$), and $d$ is the distance the source block travels between the point it was ungrasped to its final position ($\alpha_d = 1$). The task is considered solved if the final reward $R_f$ exceeds solved threshold $R_S = 5$.

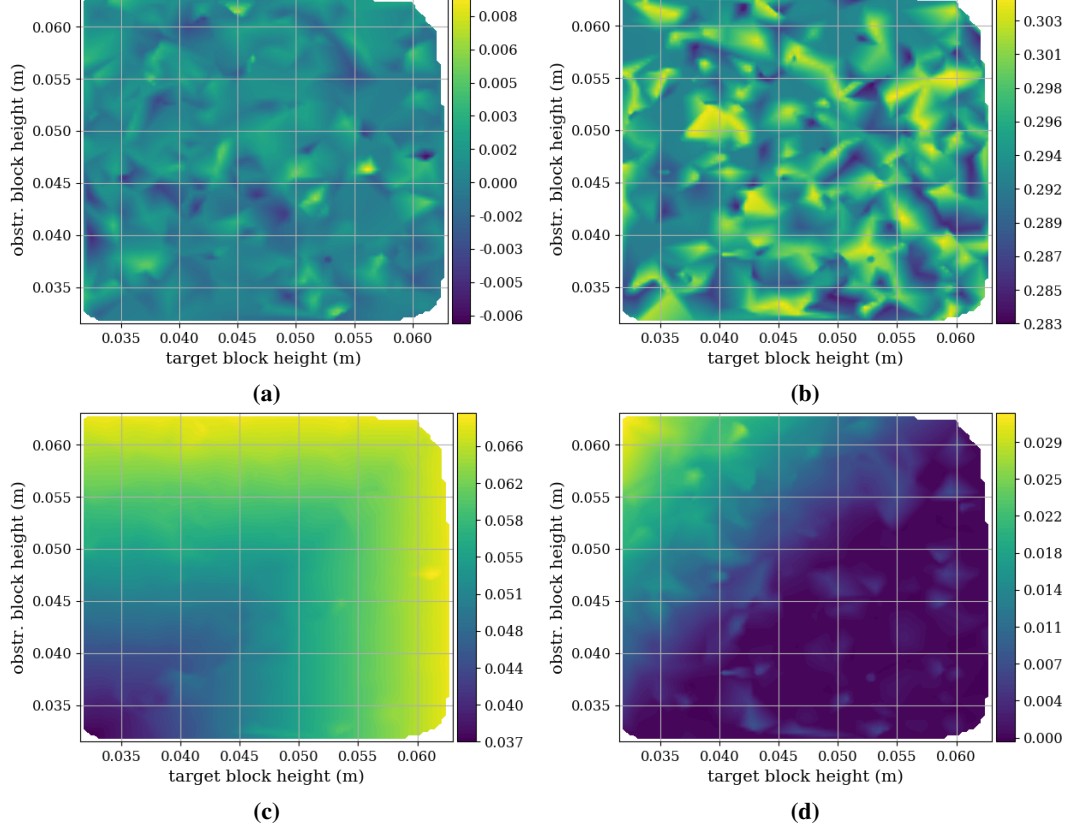

**Figure 7:** Policy parameters for the *Height-Height* experiment (shown as interpolated across the 569 dataset samples to better visualize the gradients). The units of the parameters are in meters. The parameters $\theta_{\Delta x}$ (a) and $\theta_{\Delta y}$ (b) are generally constant as they are unaffected by the variation in context variables. The notable variations occur in $\theta_{\Delta z_u}$ (c) and $\theta_{\Delta z_d}$ (d). Specifically, the relationship changes whether the obstructing block is taller or shorter than the target block (above or below the $h_t - h_o = 0$ line, respectively).

**Table 7:** Task evaluation results for using the skill library $\mathcal{K}_{HH}$ for the block stacking task. Ctrl. is the approach control (skills or one monolithic policy). Fn. Cl. is the approach's function class. Linear approaches use Bayesian ridge regression. Task Solve % is the rate of task solves over the entire context space using the approach. Methods within $\pm 2\%$ (the stochasticity of the simulator) of the best approach are bold. $|\mathbf{A}|$ is the quantity of input variables used for the approach's policy. Data is the amount of training data used for the approach. A ground truth policy is also shown, using all context variables and additional domain knowledge.

| Approach | Ctrl. | Fn. Cl. | Task Solve % | $|\mathbf{A}|$ | Data |
|---|---|---|---|---|---|
| scale-lin (ours) | 2 skills | Linear | **100.00%** (320) | 1/1 | 569 |
| monopolicy-lin-sk | 1 policy | Linear | 64.06% (205) | 2 | 569 |
| monopolicy-lin-all | 1 policy | Linear | 62.19% (199) | 2 | 581 |
| ground-truth-policy | 1 policy | Nonlin. | 100.00% (320) | * | – |

**SCALE skill selection.** In all SCALE approaches, the skills were complementary; using the entire skill library afforded greater coverage (greater task solve rate) than any single skill alone. For scale-lin, the skill selection distribution was almost even between $\mathcal{K}_1$ (43.28%) and $\mathcal{K}_2$ (56.72%), with $\mathcal{K}_3$ never being chosen. The skill $\mathcal{K}_3$ is dominated by the other two skills for this task, but $\mathcal{K}_3$ could nonetheless be useful for a different downstream task. Empirically, it was observed that $\mathcal{K}_1$ was chosen for shorter target block heights, whereas $\mathcal{K}_2$ was used elsewhere (see Fig. 8). In the nonlinear case, only $\mathcal{K}_2$ was selected.

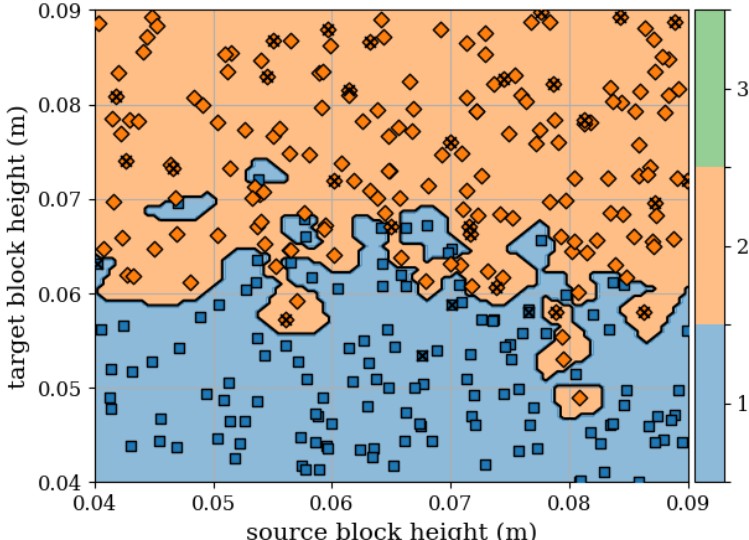

**Figure 8:** Skill selection for the scale-lin approach for the block stacking task. Skill $\mathcal{K}_1$ is generally selected when $h_2$ is short, whereas taller $h_2$ values perform better with $\mathcal{K}_2$ because $h_2 \subseteq \mathbf{A}$. Skill $\mathcal{K}_3$ is dominated by the other two skills and is not selected. Datapoints that were not solved are crossed out.

**Policy and training data ablations.** We provide additional experiments to investigate the effect of different policy functions and training data usage. The results are shown in Tab. 8, which expands Tab. 2. For the linear function class, we conduct experiments with Bayesian ridge regression (B. ridge reg.) and ordinary least squares linear regression (OLS lin. reg.). Both linear policy functions used an intercept term and were trained using unnormalized data. For the nonlinear function class, we conduct experiments with a multilayer perceptron (MLP, 16x16x16 architecture using ReLU activations) and support vector regression with a radial basis function (RBF) kernel (SVR (RBF)). The nonlinear policy functions were trained with normalized data. Additionally, we present ablations in terms of training data usage. Methods ending in "-all" use the entire batch dataset. For the full-dimensional monopolicy approaches, the "-sk" ablation uses same training data as used by the SCALE skills (340 samples). For the CREST baselines, the "-subs" ablation randomly downselects the batch dataset to the same number of samples used by SCALE (340 samples).

In general, we see that SCALE generally outperforms the full-dimensional monopolicy methods and matches the performance of the CREST baselines in most (but not all) cases. We see that increasing the amount of training data available for the baselines usually improves performance. For the linear function class, both Bayesian ridge regression and ordinary least squares linear regression produced capable approaches. For ordinary least squares linear regression, SCALE (scale-lin-ols) outperforms the full-dimensional monopolicy on a sample-adjusted basis. For the nonlinear function class, the performance of approaches was lower overall. The similarity in performance of scale-nonlin to the full-data CREST baseline is strictly due to sample size; on a sample-adjusted basis, scale-nonlin is slightly more performant. However, for support vector regression with a RBF kernel, although SCALE (scale-nonlin-svr-rbf) exceeds the performance of the full-dimensional monopolicy approaches, the CREST approaches perform more strongly (although modestly overall). Thus, we see some sensitivity for the nonlinear function class to the selection of policy function used for this task.

**Table 8:** Task evaluation results for using the skill library $\mathcal{K}_{blocks}$ for the block stacking task for a variety of policy functions and training data ablations. This table expands upon Tab. 2. Ctrl. is the approach control (skills or one monolithic policy). Fn. Cl. is the approach's function class. P. Fn. is the policy function. Task Solve % is the rate of task solves over the entire context space using the approach. Methods within $\pm 2\%$ (the stochasticity of the simulator) of the best approach are bold. $|\mathbf{A}|$ is the quantity of input variables used for the approach's policy. Data is the amount of training data used for the approach. A ground truth policy is also shown, using all context variables and additional domain knowledge. The abbreviation "mp" stands for monopolicy.

| Approach | Ctrl. | Fn. Cl. | P. Fn. | Task Solve % | $|\mathbf{A}|$ | Data |
|---|---|---|---|---|---|---|
| scale-lin (ours) | 3 skills | Linear | B. ridge reg. | **90.49%** (276) | 4/5/6 | 340 |
| monopolicy-lin-sk | 1 policy | Linear | B. ridge reg. | 80.72% (247) | 36 | 340 |
| monopolicy-lin-all | 1 policy | Linear | B. ridge reg. | 85.95% (263) | 36 | 585 |
| crest-monopolicy-lin-subs | 1 policy | Linear | B. ridge reg. | **89.87%** (275) | 5 | 340 |
| crest-monopolicy-lin-all | 1 policy | Linear | B. ridge reg. | **89.87%** (275) | 5 | 585 |
| scale-lin-ols (ours) | 3 skills | Linear | OLS lin. reg. | **90.85%** (278) | 4/5/6 | 340 |
| monopolicy-lin-ols-sk | 1 policy | Linear | OLS lin. reg. | 83.33% (255) | 36 | 340 |
| monopolicy-lin-ols-all | 1 policy | Linear | OLS lin. reg. | **90.16%** (275) | 36 | 585 |
| crest-monopolicy-lin-ols-subs | 1 policy | Linear | OLS lin. reg. | **90.52%** (277) | 5 | 340 |
| crest-monopolicy-lin-ols-all | 1 policy | Linear | OLS lin. reg. | **90.20%** (276) | 5 | 585 |
| scale-nonlin (ours) | 3 skills | Nonlin. | MLP | **63.40%** (194) | 4/5/6 | 340 |
| monopolicy-nonlin-sk | 1 policy | Nonlin. | MLP | 1.31% (4) | 36 | 340 |
| monopolicy-nonlin-all | 1 policy | Nonlin. | MLP | 10.13% (31) | 36 | 585 |
| crest-monopolicy-nonlin-subs | 1 policy | Nonlin. | MLP | 58.17% (178) | 5 | 340 |
| crest-monopolicy-nonlin-all | 1 policy | Nonlin. | MLP | **60.78%** (186) | 5 | 585 |
| scale-nonlin-svr-rbf (ours) | 3 skills | Nonlin. | SVR (RBF) | 19.61% (60) | 4/5/6 | 340 |
| monopolicy-nonlin-svr-rbf-sk | 1 policy | Nonlin. | SVR (RBF) | 1.63% (5) | 36 | 340 |
| monopolicy-nonlin-svr-rbf-all | 1 policy | Nonlin. | SVR (RBF) | 7.19% (22) | 36 | 585 |
| crest-mp-nonlin-svr-rbf-subs | 1 policy | Nonlin. | SVR (RBF) | 41.64% (127) | 5 | 340 |
| crest-mp-nonlin-svr-rbf-all | 1 policy | Nonlin. | SVR (RBF) | **56.86%** (174) | 5 | 585 |
| ground-truth-policy | 1 policy | Nonlin. | – | 95.75% (293) | * | – |

# I  Sim-to-Real Block Stacking Experiment

In this appendix, we demonstrate that the skills learned by SCALE are suitable for sim-to-real transfer. As skills are constructed using only the relevant causal variables, this is a form of *structural* sim-to-real transfer. For this experiment, we evaluate the skill library $\mathcal{K}_{blocks}$ for a real block stacking domain with a Franka Emika Panda robot manipulator (Fig. 2c). This experiment is generally similar to task evaluation in simulation, except with a smaller subset of the context space. We assess the SCALE approaches, scale-lin and scale-nonlin, against their monopolicy counterparts. We only consider the "-all" monopolicy approaches, as they were generally better performing.

## I.1  Experimental Setup

For this experiment, a smaller subset of the context space is varied, as compared to the variation across the entire context space as tested in Tab. 2. From a pool of 20 blocks, 5 were randomly chosen to be used for each experimental trial. The 20 blocks consisted of variations of 10 different colors and 2 different heights (5.7 cm or 7.6 cm). The length and width of the blocks were 4.2 cm. The 5 randomly chosen blocks were placed into the Panda robot workspace and randomly shuffled, producing variation in block $x$-position, $y$-position, and orientation. The table height $h_\pi$ was determined from manual measurement and was not varied for this experiment.

**Perception.** An Intel RealSense camera mounted to the robot wrist provided RGB-D perception of the $x$-position, $y$-position, and orientation of the blocks in the workspace. A depth observation was collected by commanding the robot above the workspace. This point cloud was then processed to yield five clusters via hidden point removal [55], RANSAC-based table plane fitting, and density-based clustering using DBSCAN [56]. Averaging the colors within each cluster yielded the block color. A least-squares optimization procedure fit a cuboid of known length and width to each cluster, yielding the position and orientation of the blocks. Block height was provided by manual input

because of inaccuracies with estimation from depth alone. The camera extrinsics were obtained via computer-aided design models of the Panda robot and wrist mount, which were confirmed via manual measurement. The camera intrinsics were used as directly reported by the camera.

**Control.** The `FrankaPy` library [57] is used to provide impedance-based control of the Panda robot.

## I.2  Experimental Results

Table 9 presents the results. For each function class, the skill library learned by SCALE outperforms the full-dimensional monopolicy baseline and is generally comparable to or slightly outcompetes the CREST monopolicy baseline. The ground truth policy matched the linear SCALE approach and is only slightly better than the nonlinear SCALE approach. Compared to the task solve rate in simulation (Tab. 2), scale-lin performed consistently, and scale-nonlin had slightly better performance. All baseline approaches generally matched their evaluation in simulation, except for monopolicy-lin-all, which had a marked degradation. This may arise from domain differences between simulation and reality. Full-dimensional approaches are more susceptible to domain shifts due to their reliance on the entire context space (all 36 variables), whereas SCALE approaches are compressed, using only a minimal subset. Error was only loosely correlated with task solve rate, and likely explains the poor performance of monopolicy-nonlin-all. Even though their errors were similar, it was observed that monopolicy-lin-all tended to underpredict the height needed to clear the target block as compared to scale-lin. This caused the target block to be pushed away from where it should have been for the goal position, leading to block stacking failures.

For both scale-lin and scale-nonlin, skill $\mathcal{K}_2$ was always chosen, as its precondition was on average greater than that of the other skills. Specifically, for scale-lin, the average preconditions were 58.88% for $\mathcal{K}_1$, 75.77% for $\mathcal{K}_2$, and 36.99% for $\mathcal{K}_3$. As the block heights used were only 5.7 cm and 7.6 cm, it is reasonable to expect that skill $\mathcal{K}_1$ would have been chosen more for shorter target block heights (per Fig. 8). For scale-nonlin, the average preconditions were $\mathcal{K}_1$: 20.17%, $\mathcal{K}_2$: 51.84%, $\mathcal{K}_3$: 1.21%.

**Table 9:** Sim-to-real evaluation results for using the skill library $\mathcal{K}_{blocks}$ for a real block stacking domain. Table columns are as described in Tab. 2. Task Solve % is the rate of successful block stacks. Error is the mean error ($\pm 1$ standard deviation) in meters between the block position when the block is ungrasped and the goal position determined at the beginning of the trial.

| Approach | Ctrl. | Fn. Cl. | Task Solve % | Error | $|\mathbf{A}|$ |
|---|---|---|---|---|---|
| scale-lin (ours) | 3 skills | Linear | **90.00%** (9) | $0.010 \pm 0.003$ | 4/5/6 |
| monopolicy-lin-all | 1 policy | Linear | 50.00% (5) | $0.008 \pm 0.003$ | 36 |
| crest-monopolicy-lin-all | 1 policy | Linear | **90.00%** (9) | $0.004 \pm 0.001$ | 5 |
| scale-nonlin (ours) | 3 skills | Nonlinear | **80.00%** (8) | $0.007 \pm 0.002$ | 4/5/6 |
| monopolicy-nonlin-all | 1 policy | Nonlinear | 10.00% (1) | $0.093 \pm 0.040$ | 36 |
| crest-monopolicy-nonlin-all | 1 policy | Nonlinear | 70.00% (7) | $0.013 \pm 0.012$ | 5 |
| ground-truth-policy | 1 policy | Nonlinear | 90.00% (9) | $0.002 \pm 0.003$ | * |

## J  Skill Library Use in a Downstream Task: Stacking a Block Tower

To demonstrate the utility of re-using skills learned by SCALE, a follow-up experiment is conducted wherein the skill library $\mathcal{K}_{blocks}$ is used for a task in which it was not specifically trained: stacking a block tower (Fig. 9). This long-horizon task can be decomposed into a number of sequential actions that must be performed correctly, so an approach that can capture the essence of a large problem and re-use smaller, modular components should perform best. Moreover, we do not perform any additional training or fine-tuning; we intentionally use the skills off-training data to test their generalization capability. This is a challenging task: in addition to the long-horizon precision involved, the skills are being evaluated increasingly out-of-distribution at each step, as the effective block heights increase beyond what is seen in training.

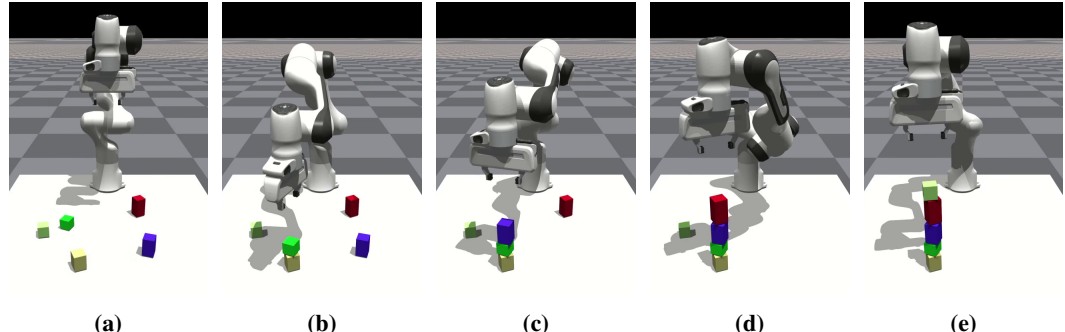

|            |            |            |            |            |
| :--------: | :--------: | :--------: | :--------: | :--------: |
| **(a)** | **(b)** | **(c)** | **(d)** | **(e)** |

**Figure 9:** The block tower task. As previously, five blocks are initially available to the robot. However, after each stack attempt, the task does not reset. Instead, the block enumeration changes, so that the previous source block becomes the new target block. This happens four times, after which the task resets. The robot must complete each of the four individual steps successfully, as failure in any step renders the entire block tower task a failure. (a) Initial task scene. (b – d) Successful block stacks for intermediate attempts. (e) A successfully stacked block tower.

**Experimental setup.** For this experiment, we assume that the robot has access to a planner and additional domain knowledge as a part of this downstream task. We assume that the robot understands that at any step, the target block should be adjusted in the following manner. First, the target block's $x$- and $y$-position should be substituted with the bottom-most block's $x$- and $y$-position. Then, the target block's height should be substituted with the sum of all heights of the previous blocks, plus a small offset (1.5 cm). Effectively, this can be seen as treating each new step as stacking upon one, increasingly taller block. We leave the development of such a planner that can provide this additional information for future work, but it suffices for this experiment that this information is available.

**Block tower results.** Table 10 shows the results for stacking the block tower. For this experiment, we use the same linear and nonlinear approaches and baselines from Sec. 6.1, including the training data ablations. Included is a ground truth policy with access to oracle information.

Overall, we see that the scale-lin approach does best for stacking a tower with five blocks, although a notable gap exists between the ground truth policy. However, a block tower success rate of 48.29% is not unreasonable, given that even the ground truth policy fails almost 30% of the time. The linear approaches are all comparable for the first stacking step, and for the second step with a $N_B = 3$ tall tower, three baseline methods slightly outperform scale-lin. However, for the last two steps, baseline approaches become markedly less performant, leading to scale-lin emerging as the best overall approach despite modest performance in an absolute sense. Each step requires successively greater extrapolation out of the training data, so an approach that can capture the smaller process well should perform best, assuming that this process also holds outside the training data. For the case of the block tower, this is generally true, so the skills learned by scale-lin are best suited for this downstream task despite the challenge of generalization to yet-unseen data.

For the nonlinear function class, performance across all approaches suffers beyond the first stacking step, where the CREST baselines outperform scale-nonlin. The challenge of extrapolation for nonlinear functions is evident here; the best linear approach for each step was better performing than any nonlinear approach (and markedly so for taller towers). Thus, out-of-distribution generalization is not observed for any nonlinear approach, whereas scale-lin exhibits modest performance in this area.

For SCALE approaches, the skill selection rate is intriguing. The skill $\mathcal{K}_1$ does not contain the target block height, which is likely why it was only selected during the first block stack attempts. However, $\mathcal{K}_2$ continues to demonstrate its robustness, as it was used for all remaining block stack attempts in the linear case and for all attempts in the nonlinear case. Its inclusion of target block height in $\mathbf{A}_{\mathcal{K}_2}$ is in fact the reason this skill can extrapolate to taller towers. Like $\mathcal{K}_2$, $\mathcal{K}_3$ also contains the block height, but this skill was generally dominated, and thus it is not surprising it was not selected.

In summary, in addition to the benefits of SCALE described previously for task learning, the capability for SCALE to learn smaller, modular skills is evident in this experiment. Although out-of-distribution generalization was not observed in the nonlinear function class, we see that in principle SCALE does offer these benefits under certain conditions, such as in the linear case. We suggest that this aspect of causal learning is often overlooked for experiments that only concern single-task learning. However, the benefits of modularity become advantageous for re-using behaviors for downstream tasks at a later time in the robot's operational lifetime.

**Table 10:** Results for re-using learned behaviors in a representative downstream task: stacking a block tower. The task solve percentage is shown for stacking a tower of at least $N_B$ blocks tall. The sequence is executed in one attempt, so a fully stacked tower ($N_B = 5$) requires 4 successful block stacking attempts. Methods within $\pm 2\%$ (the stochasticity of the simulator) of the best approach at each step are bold. For SCALE approaches, the skill selection rate at each step (not cumulative) is also shown. The abbreviation "mp" stands for monopolicy.

| Approach | | $N_B = 2$ | $N_B = 3$ | $N_B = 4$ | $N_B = 5$ |
|---|---|---|---|---|---|
| scale-lin (ours) | | **92.20%** (272) | 80.73% (222) | **65.23%** (167) | **48.29%** (113) |
| | $\mathcal{K}_1$ | 15.59% (46) | 0.00% (0) | 0.00% (0) | 0.00% (0) |
| | $\mathcal{K}_2$ | 84.07% (248) | 100.00% (275) | 100.00% (256) | 100.00% (234) |
| | $\mathcal{K}_3$ | 0.34% (1) | 0.00% (0) | 0.00% (0) | 0.00% (0) |
| monopolicy-lin-sk | | **93.22%** (275) | **87.23%** (239) | 55.08% (141) | 1.27% (3) |
| monopolicy-lin-all | | **93.56%** (276) | 76.36% (210) | 2.33% (6) | 0.00% (0) |
| crest-mp-lin-subs | | **93.20%** (274) | **85.40%** (234) | 5.84% (15) | 0.00% (0) |
| crest-mp-lin-all | | **93.92%** (278) | **85.51%** (236) | 5.84% (15) | 0.00% (0) |
| scale-nonlin (ours) | | 67.46% (199) | 2.55% (7) | 0.00% (0) | 0.00% (0) |
| | $\mathcal{K}_1$ | 0.00% (0) | 0.00% (0) | 0.00% (0) | 0.00% (0) |
| | $\mathcal{K}_2$ | 100.00% (295) | 100.00% (275) | 100.00% (256) | 100.00% (235) |
| | $\mathcal{K}_3$ | 0.00% (0) | 0.00% (0) | 0.00% (0) | 0.00% (0) |
| monopolicy-nonlin-sk | | 2.72% (8) | 0.00% (0) | 0.00% (0) | 0.00% (0) |
| monopolicy-nonlin-all | | 11.86% (35) | 0.00% (0) | 0.00% (0) | 0.00% (0) |
| crest-mp-nonlin-subs | | **84.75%** (250) | **27.37%** (75) | **0.78%** (2) | 0.00% (0) |
| crest-mp-nonlin-all | | 75.59% (223) | 11.31% (31) | 0.00% (0) | 0.00% (0) |
| ground-truth-policy | | 96.25% (282) | 90.48% (247) | 83.14% (212) | 69.96% (163) |

# K   Additional Details for Sensorless Peg-in-Hole Insertion Experiment

This appendix serves to provide greater detail for the peg insertion experiment that was described in Sec. 6.2.

**Reward function.** Our reward function consists of two terms: 1) a penalty based on the Euclidean distance of the peg from the hole, and 2) a bonus of 10 for successful insertion. We also add a regularization term based on the norm of the policy parameters. The task is considered solved if the final reward $R_f$ exceeds solved threshold $R_S = 8$.

**SCALE skill $\mathcal{K}_1$.** Unlike the other skills in $\mathcal{K}_{peg}$ that were discovered by SCALE, skill $\mathcal{K}_1$ has an empty set of relevant variables. This is surprising as it is difficult to solve this task reliably without taking the help of one of the walls, in which case the wall should show up as a relevant variable. However, we observed that $\mathcal{K}_1$ actually localizes against 2 walls instead of just 1. Hence, when SCALE intervenes on any one of the two walls, the skill is still able to complete the assembly by taking advantage of the other wall. In other words, our assumption that the context space is disentangled does not hold in this case which leads to this erroneous relevant variable set. However, the precondition would limit where this skill would be applied, as skills $\mathcal{K}_{2-5}$ are generally more performant.

**SCALE skill selection.** For scale-lin, skills $\mathcal{K}_2$ (48.44%) and $\mathcal{K}_5$ (51.56%) were chosen nearly equally. Conversely, the skill selection was more distributed for the nonlinear case: $\mathcal{K}_2$: 46.48%, $\mathcal{K}_3$: 35.16%, $\mathcal{K}_4$: 3.91%, $\mathcal{K}_5$: 14.45%. For both approaches, $\mathcal{K}_1$ was not chosen as it was dominated by the other skills.

**Table 11:** Task evaluation results for using the skill library $\mathcal{K}_{peg}$ for peg insertion for a variety of policy functions and training data ablations. This table expands upon Tab. 4. Ctrl. is the approach control (skills or one monolithic policy). Fn. Cl. is the approach's function class. P. Fn. is the policy function. Task Solve % is the rate of task solves over the entire context space using the approach. Methods within $\pm 2\%$ (the stochasticity of the simulator) of the best approach are bold. $|\mathbf{A}|$ is the quantity of input variables used for the approach's policy. Data is the amount of training data used for the approach. The abbreviation "mp" stands for monopolicy.

| Approach | Ctrl. | Fn. Cl. | P. Fn. | Task Solve % | $|\mathbf{A}|$ | Data |
|---|---|---|---|---|---|---|
| scale-lin (ours) | 5 skills | Linear | B. ridge reg. | **96.48%** (247) | 0/1/1/1/1 | 168 |
| monopolicy-lin-sk | 1 policy | Linear | B. ridge reg. | 67.19% (172) | 8 | 168 |
| monopolicy-lin-all | 1 policy | Linear | B. ridge reg. | 62.50% (160) | 8 | 210 |
| crest-monopolicy-lin-subs | 1 policy | Linear | B. ridge reg. | 66.80% (171) | 1 | 168 |
| crest-monopolicy-lin-all | 1 policy | Linear | B. ridge reg. | 62.89% (161) | 1 | 210 |
| scale-lin-ols (ours) | 5 skills | Linear | OLS lin. reg. | **96.88%** (248) | 0/1/1/1/1 | 168 |
| monopolicy-lin-ols-sk | 1 policy | Linear | OLS lin. reg. | 50.78% (130) | 8 | 168 |
| monopolicy-lin-ols-all | 1 policy | Linear | OLS lin. reg. | 67.19% (172) | 8 | 210 |
| crest-monopolicy-lin-ols-subs | 1 policy | Linear | OLS lin. reg. | 63.67% (163) | 1 | 168 |
| crest-monopolicy-lin-ols-all | 1 policy | Linear | OLS lin. reg. | 60.55% (155) | 1 | 210 |
| scale-nonlin (ours) | 5 skills | Nonlin. | MLP | **88.67%** (227) | 0/1/1/1/1 | 168 |
| monopolicy-nonlin-sk | 1 policy | Nonlin. | MLP | 18.36% (47) | 8 | 168 |
| monopolicy-nonlin-all | 1 policy | Nonlin. | MLP | 12.89% (33) | 8 | 210 |
| crest-monopolicy-nonlin-subs | 1 policy | Nonlin. | MLP | 56.64% (145) | 1 | 168 |
| crest-monopolicy-nonlin-all | 1 policy | Nonlin. | MLP | 55.47% (142) | 1 | 210 |
| scale-nonlin-svr-rbf (ours) | 5 skills | Nonlin. | SVR (RBF) | **94.53%** (242) | 0/1/1/1/1 | 168 |
| monopolicy-nonlin-svr-rbf-sk | 1 policy | Nonlin. | SVR (RBF) | 53.52% (137) | 8 | 168 |
| monopolicy-nonlin-svr-rbf-all | 1 policy | Nonlin. | SVR (RBF) | 58.20% (149) | 8 | 210 |
| crest-mp-nonlin-svr-rbf-subs | 1 policy | Nonlin. | SVR (RBF) | 57.81% (148) | 1 | 168 |
| crest-mp-nonlin-svr-rbf-all | 1 policy | Nonlin. | SVR (RBF) | 60.94% (156) | 1 | 210 |

**Policy and training data ablations.** As with the block stacking domain, we conducted experiments with several policy functions and training data ablations. Table 11 details the experimental results, which expand upon Tab. 4. In the linear function class, two policy functions were investigated: Bayesian ridge regression (B. ridge reg.) and ordinary least squares linear regression (OLS lin. reg.). An intercept term was used for both approaches, and the training data were unnormalized. In the nonlinear function class, experiments were conducted with a multilayer perceptron (MLP, 16x16x16 architecture using ReLU activations) and support vector regression with a radial basis function (RBF) kernel (SVR (RBF)). For the nonlinear policy functions, the training data were normalized. Methods with the "-all" suffix use the entire batch dataset. For the full-dimensional monopolicy approaches, the "-sk" suffix indicates that the same training data as SCALE was used (168 samples). The "-subs" suffix for the CREST baselines denotes that the batch dataset was randomly downselected to the same number of samples used by SCALE (168 samples).

Overall, we observe that the SCALE skills are highly performant across function class and policy function type. Moreover, SCALE significantly outperforms both the full-dimensional monopolicy approaches and the CREST baselines. Indeed, SCALE exceeds the performance of the baselines by around 30% for each policy function type. The success of SCALE is attributed to capturing the four modes in the data — localizing against each of the four walls — found by exploiting the underlying causal structure. The baselines, which are agnostic to such structure, do not leverage this property and are therefore limited. Unlike in the block stacking domain, we see that the effect of training data size does not necessarily yield an increase in performance for the baseline approaches.

## L   Sensorless Peg-in-Hole Insertion: Domain Shift Experiment

We evaluate the generalization capability of SCALE by evaluating it under a domain shift. All tasks are generated by uniformly sampling the relative position of the center of each wall with respect to the hole from a given range. The ranges used to generate the training and test tasks are specified in

Tab. 12. We transfer all the policies zero-shot to the test distribution. However, we do re-learn the preconditions of the scale-lin policies for the test distribution.

The evaluation results are summarized in Tab. 13. All approaches witness a sharp drop in performance. This is expected as (a) the test tasks are not guaranteed to be feasible and (b) the ranges used to generate the test task are more than double those used in training. However, our multi-skill approach scale-lin performs much better than the baselines. This highlights a key benefit of learning multiple skills. A skill may perform well on the training distribution, but it can be rendered invalid due to an unforeseen domain shift. Having a repertoire of different skills allows the robot to still complete the task by switching to a different skill. This makes our multi-skill approach more robust than single-skill approaches.

**Table 12:** Training and test distributions of the domain shift experiment in the sensorless peg-in-hole domain. The relative position of the center of each of the 4 walls is uniformly sampled from the given $(\min, \max)$ range. The ranges used to generate test tasks are more than double the ranges used to generate training tasks in the domain shift experiment. All values are in meters.

| | Train | | | | Test | | | |
|---|---|---|---|---|---|---|---|---|
| | $x$-min | $x$-max | $y$-min | $y$-max | $x$-min | $x$-max | $y$-min | $y$-max |
| Wall 1 | 0.01 | 0.05 | -0.02 | 0.02 | -0.04 | 0.10 | -0.07 | 0.07 |
| Wall 2 | -0.02 | 0.02 | -0.05 | -0.01 | -0.07 | 0.07 | -0.10 | 0.07 |
| Wall 3 | -0.02 | 0.02 | 0.01 | 0.05 | -0.07 | 0.07 | -0.04 | 0.10 |
| Wall 4 | -0.05 | -0.01 | -0.02 | 0.02 | -0.10 | -0.04 | -0.07 | 0.07 |

**Table 13:** Task evaluation results under domain shift for sensorless peg-in-hole insertion. We evaluate only linear policies as nonlinear policies perform worse in this domain. Table columns are as described in Tab. 4.

| Approach | Ctrl. | Fn. Cl. | Task Solve % | $|\mathbf{A}|$ |
|---|---|---|---|---|
| scale-lin (ours) | 5 skills | Linear | **64.84%** | 0/1/1/1/1 |
| monopolicy-lin-all | 1 policy | Linear | 44.92% | 8 |
| crest-monopolicy-lin-all | 1 policy | Linear | 39.83% | 1 |

## M   A Primer on Causality

For readers who are unfamiliar with causality, this appendix serves as a gentle "on-ramp" for understanding SCALE.

*What's a data generating process?* A data generating process (DGP) is a dynamical process that generates data in a physical system. The process is usually described by variables that characterize the system. Consider the following examples: turning a light switch on a lamp to illuminate the lightbulb; inserting a car key into an ignition and turning the starter to start a vehicle; rain showers causing rainfall. These examples can be considered data generating processes if system variables were instrumented, such as instrumenting a rain gauge to measure rainfall.

*What's a Structural Causal Model?* A Structural Causal Model (SCM) [25, 26] is a representation of a data generating process. Usually, the SCM consists of variables of a system, a graph (which is usually directed with no cycles) that describes how the variables depend on each other, and functions that describe how each variable is characterized based on that variable's causes. These functions are also called structural equations or functional equations, and each function will have its own noise variable. Noise variables (also referred to as exogenous variables) are generally jointly independent.

*What's an example of an SCM, and how can it be used?* Consider the following example of SCM $\mathfrak{C}_1$:

- $X := N_X$
- $Y := 2X + N_Y$

Here, $X$ and $Y$ are variables of our SCM, and $N_X$ and $N_Y$ are the noise terms. This SCM can also be characterized by its underlying graph, where $X \to Y$ because $X$ is a cause of $Y$. For this example, consider that $N_X$ and $N_Y$ are (independently) sampled from the uniform distribution from -10 to +10. Then, if $N_X = 2$ and $N_Y = -3$, then by the mechanics of the SCM, $X = 2$ and therefore $Y = 1$.

We now introduce the concept of an intervention, where we *set* the value of a variable to be a particular value (usually regardless of its causes or noise variables), holding all other variables equal. We can formalize this using the *do* operator [25]. Thus, an intervention $do(Y = 5)$ means that no matter what value $N_X$, $X$, or $N_Y$ take, $Y = 5$. In the previous example, under this intervention, if $N_X = 2$, then $X = 2$, but $Y = 5$ (and not 1). This type of intervention is called "hard" since it induces a structural change; other intervention types are possible, such as "soft" interventions where the functional equation of a variable changes (but not its parents).

*What's the difference between a DGP and an SCM?* In the case where the SCM captures the DGP exactly, there is no difference. However, often times we wish to learn the data generating process, and the SCM encodes the knowledge of the DGP that is currently known. In these cases, the SCM is an approximation of the underlying DGP in the physical world.

*In SCALE, what's the Data Generating Region and how does it differ from a DGP?* The Data Generating Region (DGR) introduced by SCALE provides *locality* to the data generating process. Consider a physical system where SCM $\mathfrak{C}_1$ co-exists with the following new SCM, $\mathfrak{C}_2$:

- $X := 3N_X$
- $Z := -X + N_Z$

However, it is also noticed that according to a fourth variable $A$, when $A < 0$, $\mathfrak{C}_1$ applies, whereas when $A > 0$, $\mathfrak{C}_2$ applies. The condition where these causal models apply is equivalent to how the DGR specifies *where* particular skills are defined in the context space. Note that $X, Y$, and $Z$ are not needed to define where the models apply (only $A$). A learning algorithm could use all four variables to specify where the models apply, but a minimal, compressed representation only requires one ($A$).

Moreover, $Z$ and $A$ are not needed to specify the mechanics of $\mathfrak{C}_1$ (similarly, $Y$ and $A$ for $\mathfrak{C}_2$). This is similar to how SCALE learns which variables of the context space to use for modeling the skill policy. Even though a learner could potentially use all four variables it knows about, irrelevant variables are not needed in a minimal representation.

*I'm interested in learning more about causality. Where should I start?* There are many important and useful textbooks in this area. We use Pearlian causality and SCMs as the basis of our formalism, so we recommend the reader reviewing *Causality* (Pearl 2009, 2nd edition) [25], in particular, chapters 1–3. Then, we recommend the reader reviewing *Elements of Causal Inference* (Peters, Janzing, and Schölkopf, 2017) [26], in particular, chapters 1, 3, and 6.

