# OpenReview forum: "SCALE: Causal Learning and Discovery of Robot Manipulation Skills using Simulation"
_robot-learning.org/CoRL/2023/Conference — CoRL 2023 Poster_

### Official Review · Reviewer_H8Po · 2023-07-19

**Confidence:** 4
**Originality:** Good
**Technical Quality:** Fair
**Clarity Of Presentation:** Fair
**Impact:** 3

**Recommendation:**

Weak Accept: I recommend accepting the paper, but will not argue for my recommendation if the majority of other reviewers have a different opinion.

**Review:**

While the overall intuition is relatively understandable, the details of the algorithm itself are sparse enough to the overall algorithm is difficult to follow. For example, It is not clear how the initiation set is differentiated from the initiation set. It is not clear how the policies are cleared using regression, nor what reward function is being used to train them. It is also not clear why a matrix is used to represent feature selection, instead of x_i notation, because it is unclear if a true dimensionality reduction method would be applicable to this case.
Figure 1 should help provide some of the details, but it does not really provide much beyond the same high-level overview.

The experiments provide some ambiguous results. In particular, in the block stacking task it is not clear why none of the obstacle blocks are used in any of the skills. If they are completely irrelevant to the task, then it seems to limit ---wouldn't the context specificity be useful for detecting the particular times when the distractor blocks actually inhibit motion? If not, then it is not clear why a policy composed of skills that are agnostic to these objects performs better than baselines.

By using final reward for the interventions, it seems like this could be a significant limiting factor. If a causal chain is required to achieve high reward (ex: An agent has to grab a stick to push a block to get reward), then interventions on one factor alone would give no information about the final reward.

This work could benefit from being more upfront about assumptions. Not only is the decoupled simulator assumption quite strong, but also the one-class initiation set, the regression-trained policies and the ability to find clear separations in the dataset, and the assumption f a Factorized dataset in the first place.

Some discussion of the causal relationships and the nature of the interventions would provide valuable context that this work currently lacks.

The work could benefit from additional ablatives to determine the effectiveness of some of the components: The policy regression instead of RL, the SVM one-class model, and the counterfactual testing region.


**Quality Of The Limitations Section:**

Additional details required

**Questions For Rebuttal:**

Is there a clear overview of the design decisions and ways that the algorithm is implemented in practice, along with justification for these components?

Can ablatives be added to assess these design decisions?

Is it possible to propagate reward through factors to get relevance or are skills only possible when directly connected to reward in the causal graph?

**Robotics Focus:**

Relevant but unlikely to deploy to hardware in near future

**Summary Of Paper:**

This work takes the options framework and provides an option discovery and skill-learning framework. It uses factor-specific interventions to determine which causal factors are relevant for reward in a given state. It then fits a one-class algorithm to the dimension-reduced factor-involving data to get the initiation set, regresses a policy and uses evaluation data to determine the preconditions.

**Summary Of Recommendation:**

The work introduces some interesting ideas, but the presentation is somewhat terse and the experiments do not provide sufficient evidence that this method will work generally, though the algorithm is described as being general.

---

### Official Review · Reviewer_7iCw · 2023-07-19

**Confidence:** 3
**Originality:** Very Good
**Technical Quality:** Good
**Clarity Of Presentation:** Good
**Impact:** 3

**Recommendation:**

Weak Accept: I recommend accepting the paper, but will not argue for my recommendation if the majority of other reviewers have a different opinion.

**Review:**

**Strengths**

1. **Interpretable Skills Discovery**: A major strength of the paper is its capability to discover interpretable skills.
2. **Context Space Subspace Discovery**: Another commendable feature is the ability to discover a subspace in the context space, making the framework more transferable to real-world applications.

**Weaknesses**

1. **Dependency on Prior Work**: The manuscript relies heavily on the understanding of CREST [33], making it difficult for readers not familiar with this work to grasp the concepts and significance fully. Reducing this dependency would make the paper more accessible.
2. **Lack of Comparative Discussion**: The paper falls short in discussing how the proposed method stacks up against other skill discovery methods.

**Quality Of The Limitations Section:**

Additional details required

**Questions For Rebuttal:**

1. Can you elaborate on the differences between DGR training and precondition training in skill training? Is it possible to use the trained \mathcal{D} to determine preconditions?
2. Could you provide insights on how your method compares with other self-supervised skill discovery methods such as those referenced in [22, 23]?
3. Why do you categorize this algorithm as a self-supervised learning method?
4. What is the rationale behind using the entire context space to identify D_k?
5. Is there potential to extend this work to deal with pixel observations or long-horizon tasks?

**Robotics Focus:**

Sufficient demonstration on hardware

**Summary Of Paper:**

In this paper, the authors introduce a novel method for skill discovery, which explicitly manipulates task parameters, referred to as the 'context.' This method employs simulators to observe the resultant effects of such manipulation and identify skills by the relevant subspace of context.
The authors evaluate the method's efficacy using various manipulation tasks, where it demonstrates an improved success rate in comparison to other ablation methods.
Furthermore, the authors showcase the applicability of the framework with a real-world robot.

**Summary Of Recommendation:**

The manuscript at hand builds upon previous context compression methods to extract skills in an option-oriented manner. The proposed framework, validated through multiple robotics tasks, certainly sparks interest. However, the paper falls short in providing a comprehensive discussion and comparison with other skill discovery methods.

---

### Official Review · Reviewer_zy8g · 2023-07-19

**Confidence:** 4
**Originality:** Good
**Technical Quality:** Fair
**Clarity Of Presentation:** Poor
**Impact:** 4

**Recommendation:**

Weak Accept: I recommend accepting the paper, but will not argue for my recommendation if the majority of other reviewers have a different opinion.

**Review:**


Pro:
1. Causal learning in the face of distractor features is an important and often understudied task in robotics. While most existing methods rely on hand-engineering of causal features, this paper provides a feasible way of learning such relationships automatically in simulation.
2. The method offers more interpretability to robot learning, as it is able to categorize learned policies into different skills, and pinpoint the relevant features.


Cons:
1. Clarity on main skill formulation. The main method sections (section 4 and 5) are very heavy on notations. Yet, many of the notations are only loosely defined, without a concrete example to ground them. More specifically, how are D_{A_k} and D_{D_k} computed and what are their differences? While I was able to infer them from the tables in the experiment section, I was left confused after reading section 4.1 and 4.2.
2. Clarity on figures. Would be great if the authors could elaborate on the causal feature selection section in Figure 1. More specifically, why in certain cases, the red block is kept in the final policy-relevant parameters and in other cases removed? Why is the final example failed? What is the implication of the final example? While some of these are illustrated in appendix E, it would be good to provide explanations in the main text.
3. Sample Complexity. The causal structure is learned through preforming random scene generation (section 5.1) and policy learning in all generated scenes. However, for more complicated causal structures — for instance, multi-step block stacking with complex rules — this would likely result in very  large number of scenes in order for the causal structure to be extracted. I wonder if the authors could shed any light on the sample complexity of batch data generation.
4. Visualization of the learned skills. While the experiments on learning causal policies in block stacking and peg insertion are very essential parts of explaining the main approach, the paper glosses over the resultant skills (k1-kn) in the main text, leaving it very difficult to interpret table 1 and 2 without going back and forth between the main text and the appendix.  The authors could consider moving the explanatory toy example in the main paper, and additionally, offer more insight into the individual skills in each experiment.



**Quality Of The Limitations Section:**

Limitations are addressed clearly

**Questions For Rebuttal:**

1. Can you provide more clarity on D_{A_k} and D_{D_k} in section 4?
2. Can you provide more clarity on figure 1? Currently there is no explanation in the main text on the block illustrations, making it very hard to interpret.
3. Can you comment on the sample complexity of random scene generation, and in general, how many scenes need to be generated for causal structures to be found?
4. Can you incorporate the toy example in the main text as a running example? It can serve as a running example and help ground all the notations in the method section.

**Robotics Focus:**

Sufficient demonstration on hardware

**Summary Of Paper:**

The paper proposes a method for causal learning in simulation by performing sample-based causal interventions (randomizing the environment variables) measuring task success. The learned causal relationship is captured by a dimensionality reduction matrix that filters out unimportant environment variables. Experiments validate that the method is able to extract a very compact skill library from large a number of environments.

**Summary Of Recommendation:**

Valid approach for learning causal features and improves policy interpretability. However, readability is a weakness and the paper could benefit from a rewrite. I will re-assess if the authors could address the issues in the review.

---

### Official Review · Reviewer_yTw8 · 2023-07-20

**Confidence:** 2
**Originality:** Good
**Technical Quality:** Very Good
**Clarity Of Presentation:** Good
**Impact:** 4

**Recommendation:**

Weak Accept: I recommend accepting the paper, but will not argue for my recommendation if the majority of other reviewers have a different opinion.

**Review:**

## Strengths

 - Video is informative, and was crucial for me to fully understand the approach. However it could still be improved by defining terms from CREST before using them.


## Weakness

### Accessibility to the Robot Learning Community

In its current state, the paper is not educational to those who are not fully familiar with the causal learning literature (the majority of robot learning researchers).
This is because many causal learning technical terms are used without informative examples, and assumptions deviating from the typical robot learning paper are not correctly spelled out.
Because this makes it challenging for many to understand the approach, appreciate its contributions, and grasp its limitations, people will be less likely to enter the space, extend the current frameworks, and relax existing assumptions in causal robotics.
This will create a greater divide between causality robotic researchers and the rest of the robotics learning community.
This reason is the key reason for my strong rejection.

To address this, I recommend re-writing with the goal of educating the average robot learning researcher.
I think if the paper do this in a self-contained manner and the figures are clear, many more can learn from and be excited by its contributions.
For instance, I find Figure 1 and its caption confusing because it contains symbols and uncommon words like data-generating regions and instance solvers without definitions.
Further, without watching the supplementary video, I would not have understood that the colored rectangles were blocks, not bar charts.

I understand that causality & robotics is still nascent, therefore many assumptions needs to be relaxed before it can be widely applied to robotic systems.
The limitations has listed all of these major challenges of integrating causality into robotics.
However, before causality & robotics become as mainstream as control & planning or neural networks, I think interleving these limitations in the paper writing will help the average reader more.


### Skill Definitions Depends on States

SCALE's formulation of clustering action parameters into different skills by the set of causally-relevant state variables is a interesting attempt at defining what a skill is. It is a useful and interesting application of CREST.
However, this approach relies on an interprettable, controllable state representation (an requirement inheritted from CREST's context interventions).

The current approach can't be applied to general robotic systems (e.g, deformable/fluid manipulation), and causal representation learning's goal of learning disentangled features do not imply they will be interven-able.
This means SCALE's skill learning formulation will either remain unapplicable to domains for which state-estimation is difficult or require an even stronger notion than causal disentangle feature to be achieved.

### Minor Suggestion

Many notation is introduced in 3.1 that are never used again. I recommend removing them for clarity sake. In other cases, I would recommend making sure all definitions are in 3.1. For instance, pre-conditions is used throughout the paper, yet only defined in L135.

**Quality Of The Limitations Section:**

Limitations are addressed clearly

**Questions For Rebuttal:**

- The data generation process in 5.1 remains unclear to me. It sounds either extremely expensive (e.g., requiring an inner reinforcement learning loop) or it just another way of saying its a high-level action space that can take care of the low-level heavy lifting.
 - There is nothing in the framework which constrains or assumes much about $\pi_{l}$ or $\theta$. How well would data generation perform when $\pi_{l}$ is a joint position controller (e.g. pure pursuit) and $\theta$ is the robot's joint space? Is the careful design of this action space crucial for good performance?
 - The DGR is trying to estimate the support for the policy's context distribution (L144), which can be used to filter out data relevant for policy training of that specific skill. Since the union of the DGRs are used to filter out the dataset, my interpretation is that this accounts for cases where the skill policy's context distribution support is more complicated (e.g. as in Figure 8 in the appendix)?
 - Once you train a pre-condition, it seems like DGRs are not necessary? It seems like the motivation is to ensure the data is in distribution (L175-L176), but if its not and $\pi_{uk}$ outputs bad $\theta$ then the pre-condition classifier will just give low probabilities for those contexts?

**Robotics Focus:**

Sufficient demonstration on hardware

**Summary Of Paper:**

SCALE is a representation learning algorithm for skills which are (1) generalizable, (2) account for multi-modality of robot skills. To achieve (1), they use causal feature selection to extract out a relevant subset of state variables, used for that skill policy's input. To achieve (2), they define a mode to be a set of action parameters which share the same relevant state variables from causal discovery. SCALE proposed to learn skill policies by partitioning a collected dataset by action relevant variable set and data generating region relevant variable set. The final system is a tri-level policy:
 - In the first level, a learned pre-condition classifier is used to pick one of regional compressed options.
 - In the second level, the chosen regional compressed option's upper level policy infers the low-level (third-level) policy's parameters given the compressed context.

**Summary Of Recommendation:**

The paper, in its current state, is inaccessible to people outside of causality and robotics. I think with some rewriting, many more could learn from this paper's findings.

**Update**

My main concern was the original writing and presentation was poor and inaccessible to the wider CoRL community beyond causality robotics researchers. The authors have significantly improved the writing, sufficiently addressed all my questions, and explained clearly some of the limitations/scope of their approach. I think their attempt at defining skills is interesting and insightful, although clearly limited by its dependency on an interven-able state representation. My final recommendation is a `weak accept`.

---

### Author Response · Authors · 2023-08-12
**Official Response to all Reviewers**

We would like to thank all of the reviewers for their time and feedback. We are pleased to see that the reviewers appreciated the paper’s  originality and technical quality. We appreciate that reviewers emphasized the strengths of SCALE learning interpretable skills, which our methodology provides by virtue of it being grounded in causal reasoning.

**Accessibility and clarity.** We understand that there were several issues that were raised by reviewers that involved either the paper’s accessibility to the larger robot learning community or the clarity in paper writing and explanation of our approach. We would like to use this feedback as an opportunity to improve the paper in these areas. Thus, we have uploaded a new version of the paper with the following changes to improve accessibility and clarity:
- **Improving intuition and causal terminology.** We have added greater definitions and statements providing intuition into the main paper in Sections 3.3, 4.2, and 5.1.
- **Appendix as a primer in causality.** We have dedicated an appendix for providing greater understanding of background material specifically for readers who are completely new to causality. Ideally, the above edits will allow for comprehension through only reading the main paper, but we hope the appendix will serve as a gentle “on-ramp” for robotics readers that are unfamiliar with causality. This new appendix is, for now, at the end of the appendices in Appendix K so as not to reorganize the appendix ordering during the rebuttal.
- **Greater clarity for the DGR.** To clarify several reviewer questions about the DGR (and its relationship to the precondition), we have  updated Section 4.2 to more concretely define the DGR. We believe the DGR is an important contribution of this work, as typically in the causal literature there is usually not a consideration for where the causal model applies. The DGR provides this “locality”.
- **Expanded details on CREST.** We have included more details in Section 5.1 that provide more details on CREST for readers who are unfamiliar with the work.
- **Figure 1.** To address concerns regarding Figure 1, we have moved this figure to later in the paper when we describe SCALE in Section 5, as this is when we describe the notation used in Figure 1. We have also updated the caption to help with understanding.
- **Lighten notation.** We will remove the notation of the compression being matrices, and instead refer to the associated variables using set notation. As the compression matrix and reduced set are equivalent, just having the set notation should lighten notation. Note that this change will occur following the rebuttal period (these changes are not yet made in the updated paper).

**Limitations.** We acknowledge that judicious choices of limitations are important to make progress, so that we may lay the groundwork for progress in this nascent intersection of causality and robot learning. To clarify these choices, we have also interleaved limitations and assumptions in our approach to provide this beyond what is listed in the limitations section. Examples of such clarifications are available in Sections 3.3 and 5.1.

We note that in this official response, there does not appear to be a way for us to upload the updated paper. So, we have uploaded the paper (with highlighted changes) in our initial responses to each reviewer. We note that adding these changes has temporarily increased the page limit over 8 pages; we would revise the paper again to reduce the length back to 8 pages for the camera-ready version.

---

### Decision · Program_Chairs · 2023-08-30

**Decision:**

Accept (Poster)

**Comment:**

The general consensus is that this is a good step towards an important topic (causality in robotics) with a reasonable set of experiments. However, the reviewers are generally confused about when the proposed methodology can be applied. The updated paper improved the understanding and all reviewers granted weak acceptance.

I believe the proposed method makes an explicit assumption that the world can be modeled as a *factored* MDP where a discrete set of variables (ex. gravity) is used to define the model and the intervention is done using these variables. Perhaps stating this clearly and giving examples of interventions would mitigate the issue regarding the applicability of the method.